# The fail-safe mechanism of post-transcriptional silencing of unspliced *HAC1* mRNA

**Rachael Di Santo[1], Soufiane Aboulhouda[1], David E Weinberg[1,2]\***

[1]Department of Cellular and Molecular Pharmacology, University of California, San Francisco, San Francisco, United States; [2]Sandler Faculty Fellows Program, University of California, San Francisco, San Francisco, United States

**Abstract** *HAC1* encodes a transcription factor that is the central effector of the unfolded protein response (UPR) in budding yeast. When the UPR is inactive, *HAC1* mRNA is stored as an unspliced isoform in the cytoplasm and no Hac1 protein is detectable. Intron removal is both necessary and sufficient to relieve the post-transcriptional silencing of *HAC1* mRNA, yet the precise mechanism by which the intron prevents Hac1 protein accumulation has remained elusive. Here, we show that a combination of inhibited translation initiation and accelerated protein degradation— both dependent on the intron—prevents the accumulation of Hac1 protein when the UPR is inactive. Functionally, both components of this fail-safe silencing mechanism are required to prevent ectopic production of Hac1 protein and concomitant activation of the UPR. Our results provide a mechanistic understanding of *HAC1* regulation and reveal a novel strategy for complete post-transcriptional silencing of a cytoplasmic mRNA.

## Introduction

**\*For correspondence:** david. weinberg@ucsf.edu

**Competing interests:** The authors declare that no competing interests exist.

The unfolded protein response (UPR) is a eukaryotic stress response pathway that is activated when unfolded proteins accumulate in the endoplasmic reticulum (ER) lumen (*Gardner et al., 2013*). In the budding yeast *Saccharomyces cerevisiae*, the central effector of the UPR is the transcription factor Hac1p (*Cox and Walter, 1996*; *Mori et al., 1996*; *Nikawa et al., 1996*). *HAC1* (and its metazoan ortholog *Xbp1*) is unique among eukaryotic genes in that it contains an intron that is excised through an unconventional cytoplasmic splicing reaction mediated by two proteins, Ire1p and tRNA ligase, rather than the spliceosome (*Cox and Walter, 1996*; *Kawahara et al., 1997*; *Sidrauski and Walter, 1997*; *Sidrauski et al., 1996*). In the absence of ER protein-folding stress, the intron-containing mRNA (denoted *HAC1[u]*; 'u' for UPR 'uninduced') is transcribed and exported to the cytoplasm but does not give rise to detectable protein due to the presence of the inhibitory intron (*Cox and Walter, 1996*; *Chapman and Walter, 1997*). The accumulation of unfolded proteins in the ER lumen activates the ER-resident transmembrane kinase–endonuclease Ire1p, which cleaves out the intron from *HAC1* mRNA via its cytoplasmic nuclease domain (*Sidrauski and Walter, 1997*). After the exons are joined by tRNA ligase, the resulting spliced mRNA (denoted *HAC1[i]*; 'i' for UPR 'induced') is now translated into Hac1[i]p. This active transcription factor is imported into the nucleus, where it activates the expression of UPR target genes involved in restoring protein-folding homeostasis in the ER (*Chapman et al., 1998*). Intron removal is both necessary and sufficient to relieve the post-transcriptional silencing of *HAC1* that otherwise prevents Hac1p accumulation (*Chapman and Walter, 1997*).

The post-transcriptional silencing of *HAC1* and its subsequent reversal by cytoplasmic splicing together enable a rapid UPR that does not depend on *de novo* transcription (*Rüegsegger et al.,*

**eLife digest** Molecular machines called ribosomes read the genetic instructions in an mRNA molecule and then translate them to make proteins. However, cells do not translate all of the template mRNAs that they have available into proteins; instead they have a number of ways to block the process to control when certain proteins are made.

In budding yeast, the mRNA that codes for a protein called Hac1 is always present in the cell but the protein is normally not detected. The Hac1 protein is responsible for helping the cell deal with certain types of stress, so it only accumulates when the cell is experiencing such stresses. The mRNA that encodes Hac1 (referred to as *HAC1* mRNA) contains a sequence called an intron. These sequences are normally cut out of mRNAs before they are read by the ribosome. However, the intron in the *HAC1* mRNA is unusual, because it is only removed when cells are subjected to stress. The rest of the time, this intron serves to block the production of Hac1 through a poorly understood mechanism.

Now, Di Santo et al. show the *HAC1* mRNA uses two strategies to keep itself fully repressed—both of which involve its intron. One strategy relies on a structure formed in the *HAC1* mRNA that prevents ribosomes from starting translation in the first place. However, this block is occasionally bypassed, causing some Hac1 protein to be produced when it should not be. To deal with this, the Hac1 protein that is produced contains a short protein sequence, encoded by the intron, that targets this unneeded protein for degradation. These two strategies together comprise a "fail-safe" mechanism to completely repress the *HAC1* mRNA.

Following on from these findings, it will be important to determine whether other mRNAs – both in budding yeast and in other species including humans – use a similar fail-safe strategy to block proteins from being made when they should not be.

*2001*). At the same time, a robust silencing mechanism is required to prevent ectopic accumulation of Hac1$^u$p from the abundant cytoplasmic pool of *HAC1$^u$* mRNA that might otherwise turn on UPR target genes in the absence of ER stress. The current model for silencing is that elongating ribosomes are stalled on the mRNA during translation, thereby preventing synthesis of full-length Hac1p (*Rüegsegger et al., 2001*). According to this model, the mediator of translational attenuation is a long-range base-pairing interaction between the 5′ untranslated region (UTR) and intron of *HAC1$^u$* mRNA.

The key data supporting the stalled elongation model is that the majority of *HAC1$^u$* mRNA sediments in the polysome region of a sucrose gradient (*Arava et al., 2003*; *Chapman and Walter, 1997*; *Cox and Walter, 1996*; *Kuhn et al., 2001*; *Mori et al., 2010*; *Park et al., 2011*; *Payne et al., 2008*; *Rüegsegger et al., 2001*; *Sathe et al., 2015*) despite no detectable Hac1$^u$p. Furthermore, the heavy-sedimenting *HAC1$^u$* mRNA is distributed in a discontinuous pattern with peaks and valleys that precisely match the peaks and valleys observed for polysomes (*Rüegsegger et al., 2001*). These data provide convincing evidence that heavy-sedimenting *HAC1$^u$* mRNA reflects ribosome association rather than another high-molecular-weight complex that co-sediments with polysomes, or so-called 'pseudo-polysomes' (*Thermann and Hentze, 2007*). Given this apparent ribosome association of *HAC1$^u$* mRNA, an alternative explanation for the absence of Hac1$^u$p is that Hac1$^u$p is synthesized but immediately degraded (*Cox and Walter, 1996*). However, Hac1$^u$p and Hac1$^i$p are thought to have similar half lives (*Chapman and Walter, 1997*; *Kawahara et al., 1997*), arguing against differential protein degradation as the primary mechanism that prevents Hac1$^u$p accumulation yet allows Hac1$^i$p accumulation.

Despite widespread acceptance of the stalled elongation model (*Richter and Coller, 2015*), the mechanism by which base pairing between untranslated regions causes ribosomes to stall in the open reading frame (ORF) is unknown. The reduced efficiency of translational attenuation in vitro suggested that additional factors might be involved in the inhibitory mechanism (*Rüegsegger et al., 2001*), perhaps acting to transduce the signal from the untranslated regions to translating ribosomes. However, the sequence of the base-pairing region can be changed without affecting silencing if base pairing is preserved (*Rüegsegger et al., 2001*), making it unlikely that any sequence-

specific RNA-binding proteins are involved. In addition, the more recent discovery of the No-Go Decay (NGD) pathway that recognizes stalled ribosomes and targets the associated mRNA for degradation (*Doma and Parker, 2006*) raises the question of how the ribosome–*HAC1$^u$* mRNP evades detection and subsequent turnover by the quality-control machinery. For these reasons, and others described below, we revisited the stalled elongation model, which led us to instead identify an entirely different mechanism of post-transcriptional silencing of *HAC1$^u$* mRNA that reconciles these issues.

## Results

### Very few ribosome footprints on unspliced *HAC1* mRNA

We recently reported improved ribosome-footprint profiles (*Ingolia et al., 2009*) and mRNA-abundance measurements from exponentially growing *S. cerevisiae* (*Weinberg et al., 2016*). Under these growth conditions *HAC1* mRNA is almost entirely unspliced (*Figure 1—figure supplement 1A*) and is distributed across a sucrose gradient with ~50% in the non-translating fractions and the remaining ~50% extending across all of the translating (i.e., 80*S* and larger) fractions without substantial enrichment in any particular fraction (*Figure 1A*), similar to previous observations (*Arava et al., 2003*; *Chapman and Walter, 1997*; *Cox and Walter, 1996*; *Kuhn et al., 2001*; *Mori et al., 2010*; *Park et al., 2011*; *Payne et al., 2008*; *Rüegsegger et al., 2001*; *Sathe et al., 2015*). In contrast, the well-translated actin (*ACT1*) mRNA is essentially absent from the non-translating fractions and is found mostly in large polysomes. Based on the sedimentation of *HAC1* mRNPs, we predicted that the mRNA would generate a large number of ribosome-protected fragments, in a quantity only ~2-fold fewer than similarly abundant mRNAs (based on the fraction of *HAC1* mRNA in the untranslated fractions). Strikingly, however, after normalizing for mRNA abundance *HAC1* generates the fewest ribosome-protected fragments among all expressed yeast genes (*Figure 1B*)—and ~50-fold fewer than expected from the polysome profile. Rather than providing evidence for stalled ribosomes on *HAC1$^u$* mRNA, instead these observations suggest that either ribosomes are stalled on the mRNA in a closely packed configuration that prevents nuclease cleavage between ribosomes, which would eliminate the ~28-nucleotide fragments that are sequenced in the ribosome-profiling method (*Figure 1C*, middle); or that there are not stalled ribosomes on *HAC1$^u$* mRNA (*Figure 1C*, bottom).

### Inhibited translation initiation revealed by polysome analyses

Although the polysome-like sedimentation of *HAC1$^u$* mRNA indicates ribosome association, it does not reveal if the ribosomes associated with the mRNA were ever engaged in its translation. Alternatively, the associated ribosomes might be bound in a conformation that is unrelated to translation of *HAC1$^u$* mRNA. We therefore devised an experiment to definitively determine whether the ribosomes bound to *HAC1$^u$* mRNA reflect ribosomes that were engaged in its translation. To do so, we took advantage of the observation that *HAC1* mRNA is distributed across all of the translating fractions of a sucrose gradient (*Figure 1A*). If the deep sedimentation is due to multiple translating ribosomes being stalled on a single mRNA, then reducing the number of translating ribosomes that can fit on the mRNA should shift the sedimentation pattern toward lighter fractions. We designed a series of constructs containing point mutations in the first exon of *HAC1* that created premature termination codons, which reduce the size of the ORF and thereby limit the number of translating ribosomes (*Figure 1D*, top). To ensure that the mutant alleles were expressed at near wild-type levels, we replaced the endogenous *HAC1* allele without disrupting flanking regulatory regions. Remarkably, each of the mutant mRNAs had a sedimentation pattern that was indistinguishable from the wild-type mRNA (*Figure 1D*, bottom). In the most extreme case, the mRNA containing a 21-nucleotide ORF that can only accommodate a single translating ribosome still co-sedimented with polysomes containing upwards of 10 ribosomes (fraction 14 of the gradient). These data provide direct evidence that the polysome-like sedimentation of *HAC1$^u$* mRNA is not due to stalled ribosomes on the mRNA.

We hypothesized that the ribosome association of *HAC1$^u$* mRNA was instead due to non-specific interactions between the mRNA and bona fide polysomes formed on other mRNAs. To evaluate the extent of such non-specific interactions, we introduced an exogenous control RNA that should not

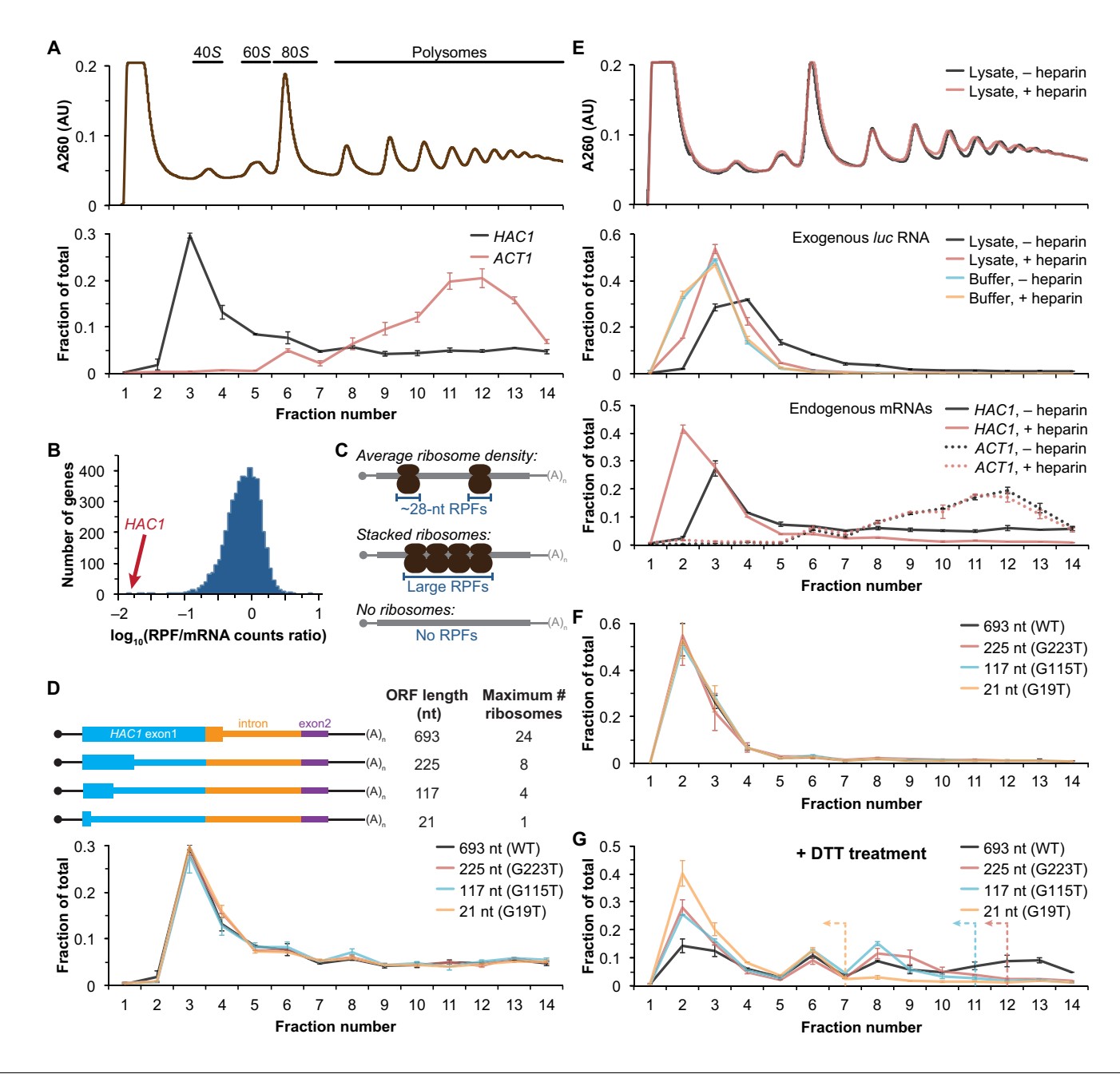

**Figure 1.** Ribosome density on unspliced *HAC1* mRNA. (**A**) Polysome analysis of *HAC1* and *ACT1* mRNAs. Extracts prepared from exponentially growing yeast cells were fractionated on 10–50% sucrose gradients, with absorbance at 260 nm monitored (top). The relative distributions of *HAC1* and *ACT1* mRNAs across fractions were determined by qRT-PCR (bottom). Shown are the mean ± SEM with *n* = 2 (i.e., the range), expressed as a fraction of the total mRNA detected. (**B**) Histogram of ribosome densities measured by ribosome profiling and RNA-seq. The ratio of the number of ribosome-protected fragments (RPFs) to the number of RNA-seq reads (mRNA counts) was calculated for each of 4838 expressed yeast genes (data from *Weinberg et al., 2016*). Shown is the distribution of log-transformed ratios in bins of 0.05, with the position of *HAC1* indicated. (**C**) Possible scenarios to explain a lack of RPFs. While an mRNA with average ribosome density will generate many ~28 nucleotide (nt) RPFs (top), the close packing of stacked ribosomes could inhibit the RNase digestion between ribosomes required to generate ~28 nt RPFs (middle). Alternatively, an mRNA that does not contain translating ribosomes would not generate RPFs (bottom). (**D**) Polysome analysis of *HAC1* mRNA variants with shortened ORFs. G-to-T point mutations were introduced into the first exon of *HAC1* to generate premature stop codons, with the resulting ORFs shown as thick colored boxes (constitutive 5′- and 3′-UTRs located within exons 1 and 2, respectively, are shown as thin black lines; other untranslated regions are depicted as thin colored boxes; and the coding regions of exons 1 (teal) and 2 (purple) are labeled as 'HAC1 exon1' and 'exon2', respectively). The maximum number of ribosomes that could be accommodated was calculated based on each ribosome occupying 28 nt. Polysome analysis was performed as in (**A**), with

*Figure 1 continued on next page*

*Figure 1 continued*

data for wild-type *HAC1* from (**A**) duplicated for comparison. (**E**) Effects of heparin on polysome analysis. Purified uncapped *luciferase* (*luc*) RNA was added to either lysate or lysis buffer in the absence (–) or presence (+) of 0.2 mg/ml heparin. Polysome analysis was performed as in (**A**) with absorbance at 260 nm monitored (top), and the relative distributions of exogenous *luc* RNA (middle) and endogenous *HAC1* and *ACT1* mRNAs (bottom; in lysate only) were determined. (**F**) Refined polysome analysis of *HAC1* mRNAs. Extracts were prepared in heparin-containing lysis buffer from strains shown in (**D**). Polysome analysis was performed as in (**A**). (**G**) Polysome analysis of *HAC1* mRNAs during the UPR. Strains shown in (**D**) were grown to mid-log phase and treated with 8 mM DTT for 20 min before harvesting. Extracts were prepared in heparin-containing lysis buffer, and polysome analysis was performed as in (**A**). Dotted lines indicate the fractions after which the corresponding color-coded mutant mRNAs would not be expected to sediment based on ORF length.

The following figure supplement is available for figure 1:

**Figure supplement 1.** Splicing of *HAC1* mRNA.

be translated: an uncapped luciferase-encoding RNA purified from an in vitro transcription reaction. We analyzed the sedimentation behavior of this control RNA when it was added to lysis buffer compared to when it was added to the yeast lysate prior to centrifugation. Surprisingly, some of the exogenous RNA was found in the translating fractions of the lysate—a behavior not observed in lysis buffer alone (*Figure 1E*, middle). Through extensive optimization, we found that the addition of heparin to the lysis buffer (at a concentration of 0.2 mg/ml) was sufficient to largely prevent the deep sedimentation of exogenous RNA. The addition of heparin also had a major effect on the sedimentation of *HAC1* mRNA, as most (83%) now sedimented in the non-translating fractions of the gradient (*Figure 1E*, bottom). In contrast, the sedimentation of *ACT1* mRNA (*Figure 1E*, bottom) and the overall polysome profile (*Figure 1E*, top) were largely unchanged, suggesting that heparin competed away non-specific interactions without disrupting bona fide polysomes.

Based on these results, we re-analyzed the *HAC1* mutants with shortened ORFs using heparin-containing lysis buffer. Under these conditions, the polysome co-sedimentation of each of the mRNAs was greatly reduced (from ~50% to < 20%) but there was still no difference among the constructs (*Figure 1F*), providing further evidence against stalled ribosomes on *HAC1*$^u$ mRNA. Importantly, when we treated cells with the reducing agent dithiothreitol (DTT) to induce the UPR and concomitant splicing of *HAC1* mRNA (*Figure 1—figure supplement 1C*) and repeated the experiment, the variant mRNAs now displayed the expected differential sedimentation based on ORF length (*Figure 1G*), validating our experimental design. Together, these results demonstrate that at steady state *HAC1*$^u$ mRNA is not associated with either actively elongating or stalled ribosomes. This indicates that the primary block to production of Hac1$^u$p is at the stage of translation initiation, not translation elongation as previously proposed (*Chapman and Walter, 1997*; *Richter and Coller, 2015*; *Rüegsegger et al., 2001*).

## An additional silencing mechanism downstream of translation initiation

The absence of ribosomes on *HAC1*$^u$ mRNA suggests that translation initiation is inhibited. To further understand the mechanism of this inhibition, we used a green fluorescent protein (GFP) reporter system that has been previously shown to recapitulate post-transcriptional silencing by the *HAC1* intron (*Chapman and Walter, 1997*; *Rüegsegger et al., 2001*). We replaced the first exon of *HAC1* with the GFP ORF lacking its own stop codon (*Figure 2B*), which allowed us to quantitatively analyze post-transcriptional silencing of GFP using a combination of flow cytometry (protein abundance), quantitative RT-PCR (mRNA abundance), and sucrose gradient fractionation (ribosome density). When *GFP* was embedded in an otherwise wild-type *HAC1* context the mRNA sedimented almost entirely in the non-translating fractions (*Figure 2A*), and there was no detectable fluorescence above background (*Figure 2C*). In contrast, cells expressing a reporter construct missing the entire intron displayed a strong fluorescence signal (*Figure 2C*), and most of the mRNA was found in the translating fractions (*Figure 2A*). Thus, the reporter *GFP* mRNA behaves similarly to endogenous *HAC1* mRNA.

To determine whether 5′-UTR–intron base pairing is required to prevent ribosome loading on the reporter mRNA, we designed constructs in which the sequence implicated in base-pairing interactions in either the 5′-UTR or the intron was mutated to its complement to disrupt the interaction

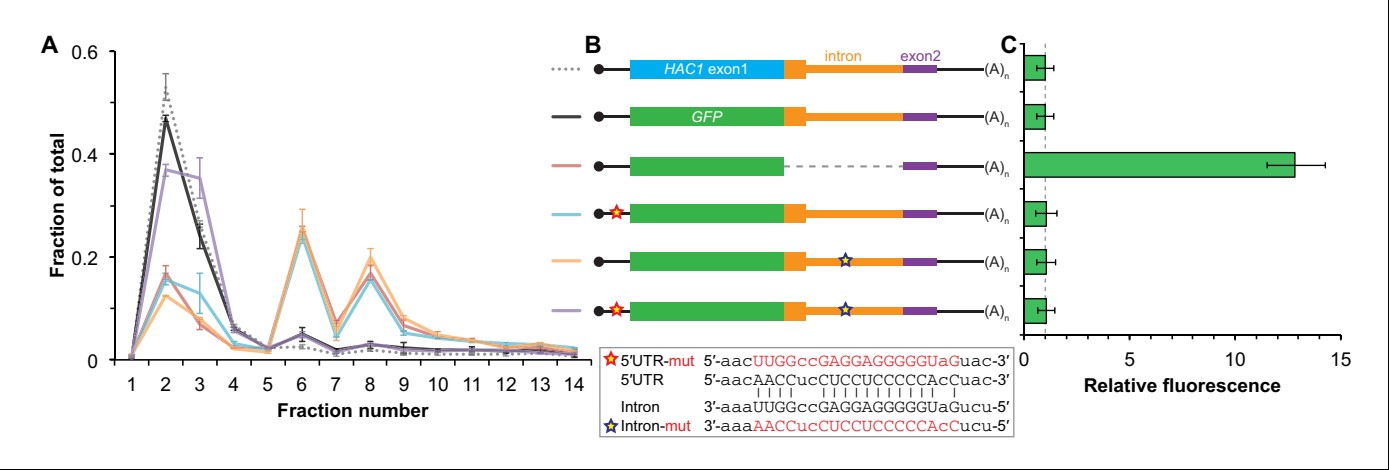

**Figure 2.** Contribution of long-range base pairing to intron-dependent silencing. (**A**) Polysome analysis of reporter mRNAs. Extracts were prepared in heparin-containing lysis buffer from strains expressing the *GFP* reporter mRNAs depicted in (**B**). Polysome analysis was performed as in *Figure 1A*, with data for wild-type *HAC1* from *Figure 1F* duplicated for comparison. (**B**) Design of reporter mRNAs. Constructs are depicted as in *Figure 1D*, with the dotted line indicating a deleted region. Colored stars indicate mutations to the base-pairing region, with specific nucleotide changes shown below in red. (**C**) Flow cytometry analysis of reporter strains. Strains expressing the *GFP* reporter mRNAs depicted in (**B**) were grown to mid-log phase and analyzed by flow cytometry. Plotted is the median GFP intensity (normalized to cell size) of the cell population relative to background fluorescence in the wild-type (no GFP) strain with error bars indicating quartiles of the cell population, all averaged across replicates (*n* = 2–7).

The following figure supplement is available for figure 2:

**Figure supplement 1.** Characterization of *GFP* reporter strains.

(*Figure 2B*, bottom), as was done previously (*Rüegsegger et al., 2001*). Both mutant mRNAs sedimented mostly in the translating fractions in a manner that was similar to the intronless construct (*Figure 2A*). When we combined the 5′-UTR and intron mutations and thereby restored base pairing, the mRNA was now found almost exclusively in the non-translating fractions and resembled the original reporter mRNA. These results demonstrate that base pairing between the 5′-UTR and intron is required to prevent ribosome loading by directly impeding the binding or progress of the scanning ribosome.

Given that the base-pairing mutant mRNAs were loaded with ribosomes similarly to the intronless mRNA (*Figure 2A*), we expected to observe GFP expression from the mutant mRNAs that was similar to that of the intronless construct. Remarkably, however, neither of the strains expressing a base-pairing mutant mRNA had any GFP detectable by either flow cytometry (*Figure 2C*) or immunoblotting (*Figure 2—figure supplement 1*). The lack of GFP signal despite polysome sedimentation was not due to low mRNA abundance, as the mutant mRNAs were present at similar levels compared to the intronless mRNA (*Figure 2—figure supplement 1A*, compare construct 3 with constructs 5–6). These results suggest that an additional silencing mechanism acting downstream of translation initiation prevents GFP accumulation when base pairing is disrupted.

## Post-translational silencing by the intron-encoded C-terminal tail

In addition to removing the intron portion of the base-pairing interaction, splicing of *HAC1* mRNA also alters the C-terminal tail of the encoded protein: The ORF in *HAC1ᵘ* mRNA has a 10-amino-acid tail encoded by the intron, which in *HAC1ⁱ* mRNA is replaced by an 18-amino-acid tail encoded by the second exon (*Figure 3A*). The fact that the polypeptide tails encoded by the *HAC1ᵘ* and *HAC1ⁱ* mRNAs are different suggests that the 10-amino-acid tail unique to Hac1ᵘp may be functionally important. We therefore hypothesized that the intron-encoded C-terminal tail may be involved in the additional silencing mechanism revealed by our base-pairing mutant reporters (*Figure 2*).

To investigate this possibility, we designed a reporter construct in which we removed the entire intron except for the 5′ end that codes for the 10-amino-acid tail of Hac1ᵘp (*Figure 3B*, fourth construct). Remarkably, cells expressing this construct had no detectable GFP (*Figure 3C*), despite the

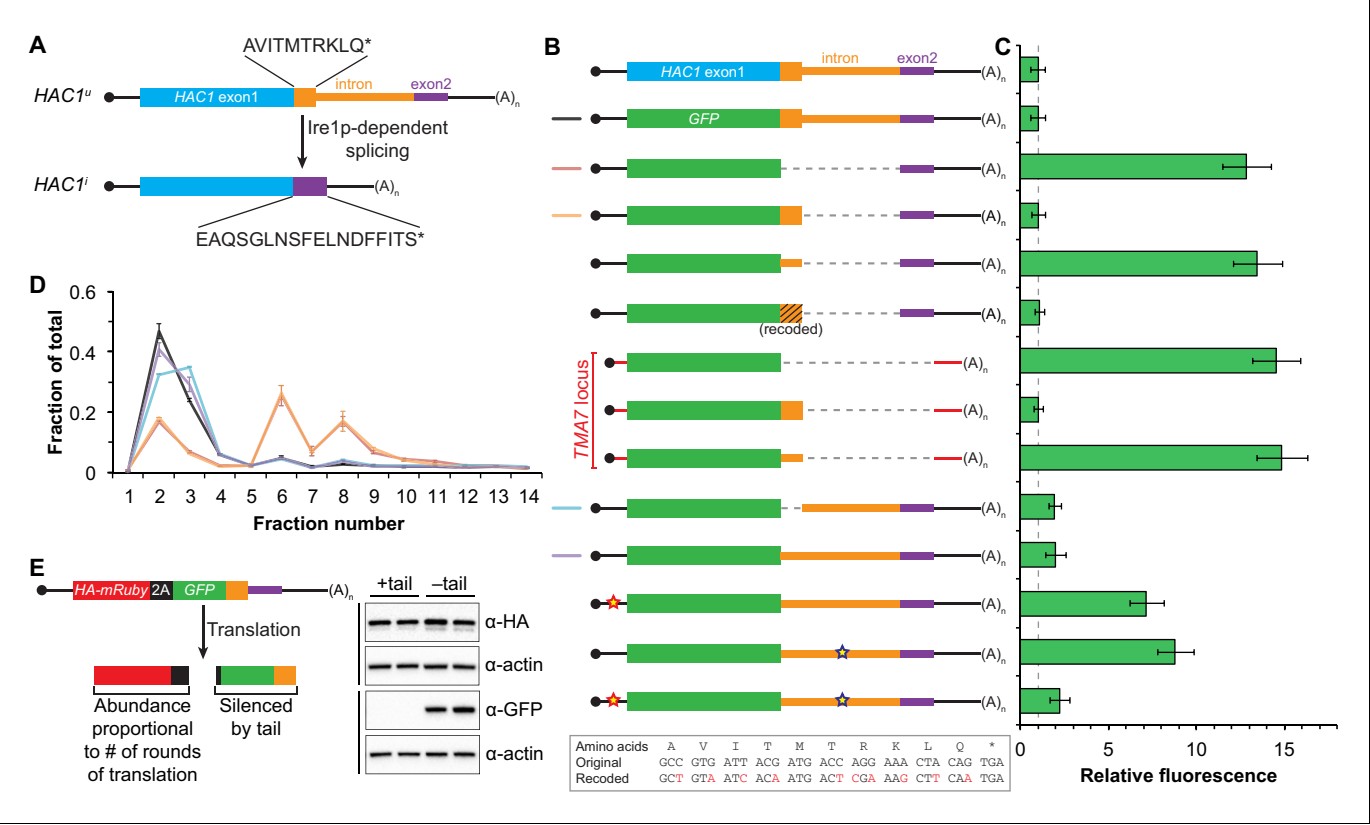

**Figure 3.** Post-translational silencing mediated by the intron-encoded C-terminal tail. (**A**) Schematic of *HAC1* mRNA splicing. The proteins encoded by *HAC1u* and *HAC1i* mRNAs differ in their C-terminal tails, with the amino acid sequences indicated. (**B**) Design of reporter mRNAs. Black shading indicates recoding, with the original and recoded sequences depicted below (mutations in red). Untranslated regions colored red correspond to those of the *TMA7* mRNA, with the reporter gene integrated at the *TMA7* rather than *HAC1* locus. Otherwise constructs are depicted as in *Figure 2B*. (**C**) Flow cytometry analysis of reporter strains. Strains expressing the *GFP* reporter mRNAs depicted in (**B**) were analyzed as in *Figure 2C*, with data for the first three strains duplicated from *Figure 2B* for comparison. (**D**) Polysome analysis of reporter mRNAs. Extracts were prepared in heparin-containing lysis buffer from strains expressing the *GFP* reporter mRNAs indicated in (**B**). Polysome analysis was performed as in *Figure 1A*, with data for the wild-type and intronless *GFP* reporters from *Figure 2A* duplicated for comparison. (**E**) Differentiating between co-translational and post-translational silencing mechanisms. Left: Schematic of reporter construct that generates two separate polypeptides from each round of translation. Right: Extracts were prepared from strains expressing reporter mRNAs that either encoded the 10-amino-acid C-terminal tail of Hac1up (+tail) or contained a stop codon just before the tail (–tail). Immunoblotting was used to detect HA-tagged mRuby (top) and GFP (bottom), with actin as a loading control. Two biological replicates are shown for each genotype.

The following figure supplements are available for figure 3:

**Figure supplement 1.** Additional analyses of *GFP* reporter constructs.

**Figure supplement 2.** Identifying a 2A peptide sequence that is active in *S. cerevisiae*.

corresponding mRNA being abundant (*Figure 3—figure supplement 1A*) and detected on polysomes due to the absence of the base-pairing interaction (*Figure 3D*). Thus, the coding region at the 5′ end of the intron is sufficient for complete silencing of the GFP reporter. Introducing a termination codon between *GFP* and the intron restored robust fluorescence comparable to that observed for the intronless construct, which implicated translation of the 5′ end of the intron as required for silencing. Furthermore, making 10 nucleotide changes that maintained the coding potential of the intron-encoded tail (*Figure 3B*, bottom) had no effect on silencing (*Figure 3C*), suggesting that the amino-acid sequence encoded by the 5′ end of the intron is more important than the nucleotide sequence itself.

To determine whether the 10-amino-acid element alone was sufficient for silencing in the absence of any other *HAC1* sequences, we expressed *GFP* from a different locus in the yeast genome (*TMA7*) either with or without the 10-amino-acid tail. When the 10-amino-acid sequence was either absent or not translated due to a premature stop codon, we observed robust GFP signal (*Figure 3C*). In contrast, there was no fluorescence detected in cells expressing GFP containing the 10-amino-acid tail. Thus, the C-terminal tail of Hac1$^u$p is sufficient for silencing independently of the rest of the *HAC1* intron and any other *HAC1* sequences.

Having established the effects of the 10-amino-acid tail in isolation, we next examined the effects of the tail when translation initiation was inhibited by the base-pairing interaction. In an otherwise wild-type *HAC1* context, preventing translation of the 10-amino-acid tail either by deleting the entire sequence or by introducing a stop codon caused GFP to accumulate to low but detectable levels (*Figure 3C*) without greatly affecting the sedimentation of the corresponding mRNAs (*Figure 3D*). Thus, even when base pairing is intact there is some low-level accumulation of GFP that is normally suppressed by the 10-amino-acid silencing element.

In the context of reporter constructs lacking the intron-encoded C-terminal tail, mutations in either the 5′-UTR or intron that disrupted base pairing greatly increased the amount of GFP, while the compensatory double-mutant construct with restored base pairing had only low levels of GFP (*Figure 3C* and *Figure 3—figure supplement 1B*). These results are in agreement with previous GFP reporter experiments, which used constructs that contained a stop codon between the *GFP* and intron sequences and therefore had inadvertently eliminated the effects of the 10-amino-acid tail (*Chapman and Walter, 1997*; *Rüegsegger et al., 2001*).

Together, our GFP reporter experiments indicate that the *HAC1* intron mediates post-transcriptional silencing through a pair of independent but partially redundant mechanisms: base-pairing interactions with the 5′-UTR that inhibit translation initiation, and a novel silencing mechanism mediated by the intron-encoded C-terminal tail of Hac1$^u$p. Robust expression requires that both silencing mechanisms be inactivated, as would happen simultaneously when the intron is removed by Ire1p-dependent splicing.

What is the mechanism by which the intron-encoded tail of Hac1$^u$p silences gene expression? Because disrupting translation of the tail elevated protein levels (*Figure 3C*) without affecting polysome formation (*Figure 3D*), we inferred that the tail was exerting its effect downstream of translation initiation. We therefore reasoned that the 10-amino-acid sequence was acting either by halting translation across the entire mRNA (after polysome formation); or by promoting protein degradation. Our inability to detect GFP containing the 10-amino-acid tail (*Figure 3C*) prevented us from directly comparing the half lives of GFP with and without the tail. Instead, to distinguish between stalled elongation and protein degradation, we designed a reporter construct that generates two separate polypeptides from a single round of translation through co-translational 'cleavage' mediated by a viral 2A peptide (Sharma et al., 2012). After screening for a 2A peptide sequence that functions efficiently in *S. cerevisiae* (*Figure 3—figure supplement 2*), we designed a construct containing HA-mRuby and GFP sequences separated by the P2A peptide (derived from porcine teschovirus-1). The GFP sequence downstream of P2A was appended with the 10-amino-acid tail (or was not, as a control), which should lead to the absence of detectable GFP regardless of the mechanism of action. In contrast, the upstream HA-tagged mRuby should only be detected if translation itself is not affected by the 10-amino-acid sequence, since it reports on the number of rounds of translation but is not covalently linked to the inhibitory tail. Assaying for GFP by immunoblotting revealed that accumulation of the protein was suppressed by the 10-amino-acid tail, as expected (*Figure 3E*). In contrast, HA-mRuby was detected at similar levels whether or not the tail was included in the construct. These results indicate that the tail functions downstream of translation, likely by acting as a 'degron' (*Varshavsky, 1991*) that targets the protein for immediate degradation after synthesis (*Cox and Walter, 1996*).

## Identification of *DUH1* through a genetic selection

If the C-terminal tail of Hac1$^u$p functions as a degron, additional proteins may be involved in recognizing the degron and targeting the covalently linked protein for degradation. To identify such *trans*-acting factors, we used a genetic approach that took advantage of the strong silencing phenotype imparted by the 10-amino-acid tail alone (*Figure 3C*). We constructed strains in which the first exon of *HAC1* was replaced by *HIS3*, which we reasoned might behave like the analogous *GFP* reporter

genes and be completely silenced by the 10-amino-acid sequence. Similarly to *GFP*, replacing the first exon of *HAC1* with *HIS3* but keeping the *HAC1* locus otherwise intact prevented expression of His3p as evidenced by histidine auxotrophy (*Figure 4A*). Removing the entire intron restored His3p expression and growth of the corresponding strain on medium lacking histidine, suggesting that silencing was mediated by the *HAC1* intron. Strains containing a stop codon between *HIS3* and the intron (to prevent translation of the 10-amino-acid degron) had a low but detectable level of growth on histidine-lacking medium (*Figure 4A*), consistent with the weak fluorescence signal observed from the corresponding *GFP* reporter construct (*Figure 3C*). On the other hand, strains containing His3p appended with the C-terminal tail of Hac1ᵘp but no other elements of the *HAC1* intron could not grow on medium lacking histidine. These results indicate that the degron is sufficient for functional silencing of *HIS3* expression, providing a useful genetic tool to identify additional genes involved in the degron-dependent silencing mechanism.

Although we had initially intended to use chemical mutagenesis to generate silencing-defective mutants, upon streaking out the selection strain on histidine-lacking medium we noticed that a small number of slow-growing colonies appeared even without mutagen treatment. We reasoned that such spontaneous suppressor strains would have very few mutations, making it possible to identify suppressor mutations by whole-genome sequencing without requiring backcrossing or forming complementation groups. Thus, to isolate mutants with defective degron-dependent silencing ('*dds* mutants') we simply plated the selection strain expressing HA-tagged His3p with the 10-amino-acid tail on histidine-lacking medium and isolated the rare single colonies that grew for further analysis (*Figure 4B*). From five independent platings we isolated a total of 123 mutant strains that, when re-streaked, could grow on medium lacking histidine. Sanger sequencing of the C-terminal region of the reporter gene in each mutant identified 15 strains harboring a *cis* mutation that either altered the sequence of the degron, introduced a premature stop codon before or within the degron, or removed the stop codon of the degron resulting in a six-amino-acid C-terminal extension (*Figure 4—figure supplement 1A*)—all of which provided confirmation that the genetic selection worked as desired.

From the strains that displayed histidine prototrophy, we picked 35 (including four *cis* mutants) to analyze by immunoblotting and found that all had detectable HA-His3p (*Figure 4—figure supplement 1B*). We selected 20 of these strains with unknown mutations for whole-genome sequencing (as well as the parental selection strain as a reference, and three strains with known mutations in the degron as positive controls for our variant-calling procedure), taking care to select strains that varied widely in HA-His3p abundance or growth rate to minimize our chances of sequencing the same mutation in multiple strains. We sequenced the 24 genomes together in a single lane of a HiSeq sequencer using 50-nucleotide single-end reads, which provided 38–74X coverage (9.3–18.0 million reads) of each yeast genome. We then used standard mapping and variant-calling tools (BWA and FreeBayes, respectively) to identify variants that were absent from the parental genome, which successfully recovered the positive-control *cis* mutants. Remarkably, 17 out of the 20 strains that we sequenced contained a mutation in the same gene *YJL149W* (*Figure 4C* and *Figure 4—figure supplement 1C*), and in each case we confirmed the mutation by Sanger sequencing (*Figure 4—figure supplement 1D*). Because no other ORF was found mutated in more than one of the 20 mutant strains (*Figure 4—figure supplement 1C*), we focused our follow-up efforts on *YJL149W*.

*YJL149W* had previously been named *DAS1* for 'D̲st1-delta 6-A̲zauracil S̲ensitivity 1̲' when it was identified in a genetic screen unrelated to the UPR (*Gómez-Herreros et al., 2012*). Based on the protein's domain structure (containing an F-box domain and leucine-rich repeats) and physical interactions with the SCF core components Cdc53p and Skp1p (*Willems et al., 1999*), *YJL149W* was annotated as a 'putative SCF ubiquitin ligase F-box protein' (*Cherry et al., 2012*). F-box proteins act as adapters to target substrates for ubiquitination and subsequent degradation by the proteasome (*Skaar et al., 2013*). We therefore propose to rename this gene *DUH1* for 'D̲egrader of U̲nspliced H̲AC1 gene product 1̲' to reflect its role in the degradation of Hac1ᵘp, as we demonstrate later.

The *DUH1* variants that we identified in our 17 strains comprised 16 different mutations, of which 8 were nonsense, 7 were missense, and 1 was a frameshift (*Figure 4C*, right). The nonsense mutations tended to cluster in the first half of the ORF, suggesting that they likely functioned as null alleles. In addition, three of the 17 strains containing mutations in *DUH1* did not contain any other mutations, implicating the *DUH1* mutations as causative for the phenotype. We therefore tested whether knocking-out *DUH1* (which is non-essential) in the original selection strain recapitulated the

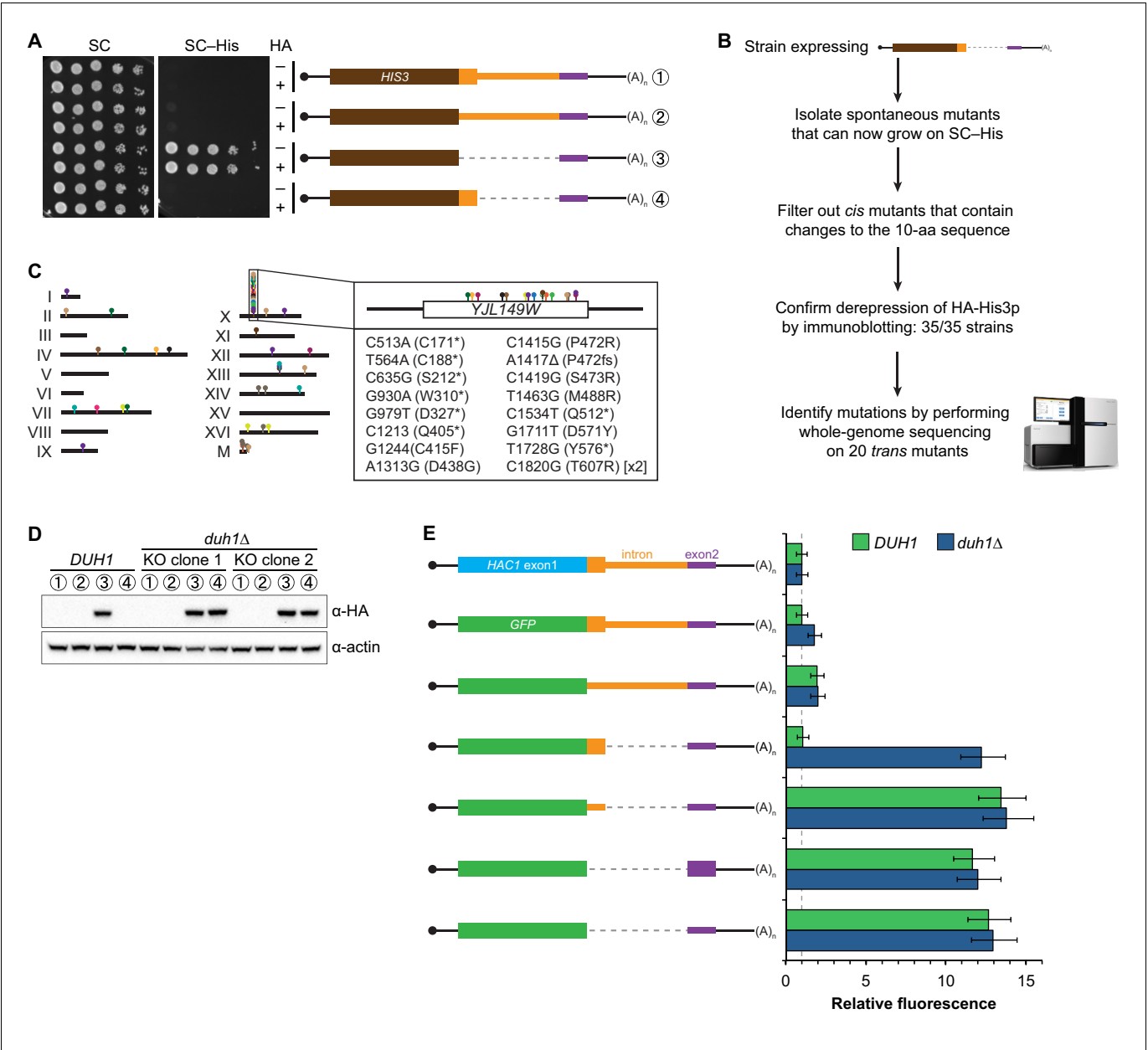

**Figure 4.** Identification of *DUH1* through a genetic selection. (**A**) Evaluating a genetic reporter for intron-dependent silencing. Strains expressing the indicated *HAC1*-based *HIS3* reporter mRNAs (depicted as in *Figure 2B*) either without (–) or with (+) an N-terminal HA tag were grown to saturation, and 10-fold dilution series were plated on either SC or SC–His media. (**B**) Flowchart of genetic selection for *dds* mutants. After selecting for spontaneous mutants that could grow on medium lacking histidine, restreaked clones were filtered out for *cis* mutants, verified to be expressing HA-His3p, and a subset analyzed by whole-genome sequencing. (**C**) Chromosome map of mutations identified by whole-genome sequencing. Each color corresponds to a different *dds* mutant strain. Shown on the right is the *DUH1* locus (*YJL149W*), with locations of nucleotide changes (with respect to *DUH1* start codon) and corresponding amino changes listed below. (**D**) Effect of *DUH1* disruption on expression of the genetic reporter. Strains expressing the indicated reporter mRNAs depicted in (**A**) in either a *DUH1* or *duh1Δ* (two independent clones) background were grown to mid-log phase. Extracts were prepared and immunoblotted for HA-His3p and actin loading control. (**E**) Flow cytometry analysis of reporter strains. Strains expressing the indicated *GFP* reporter mRNAs in either a *DUH1* or *duh1Δ* background were analyzed as in *Figure 2C*. Data is plotted as in *Figure 2C* (*n* = 2 for all strains).

The following figure supplements are available for figure 4:

**Figure supplement 1.** Results and validation of the genetic selection.

**Figure supplement 2.** Additional analysis of *GFP* reporter constructs.

de-silencing phenotype. Disrupting *DUH1* led to a dramatic increase in the steady-state abundance of HA-His3p containing the 10-amino-acid tail, while having no effect on HA-His3p lacking the tail or constructs repressed by long-range base pairing (*Figure 4D*). Analogously, deleting *DUH1* in the GFP reporter strains completely eliminated the silencing effect of the Hac1^up tail but had no effect on base-pairing-mediated silencing (*Figure 4E* and *Figure 4—figure supplement 2*). Notably, in the absence of *DUH1* the GFP reporter in a wild-type *HAC1* context was now expressed at the same leaky level as previously seen for the reporter in which a stop codon was positioned between *GFP* and the intron, indicating that degron-dependent silencing is required to suppress leaky GFP expression. Together, these results provide genetic evidence that *DUH1* is the adapter protein that recognizes the Hac1^up tail and targets the covalently attached protein for degradation.

## Effects of *DUH1* on Hac1p abundance, synthesis, and turnover

Having established that *DUH1* is required for degron-dependent silencing of two different reporter genes, we returned to *HAC1* itself. To detect Hac1p we introduced a 3xHA tag at the extreme N terminus of the endogenous *HAC1* allele (*Figure 5A*), which did not interfere with its post-transcriptional silencing in the absence of the UPR (*Figure 5B and C*). To determine which Hac1p isoforms

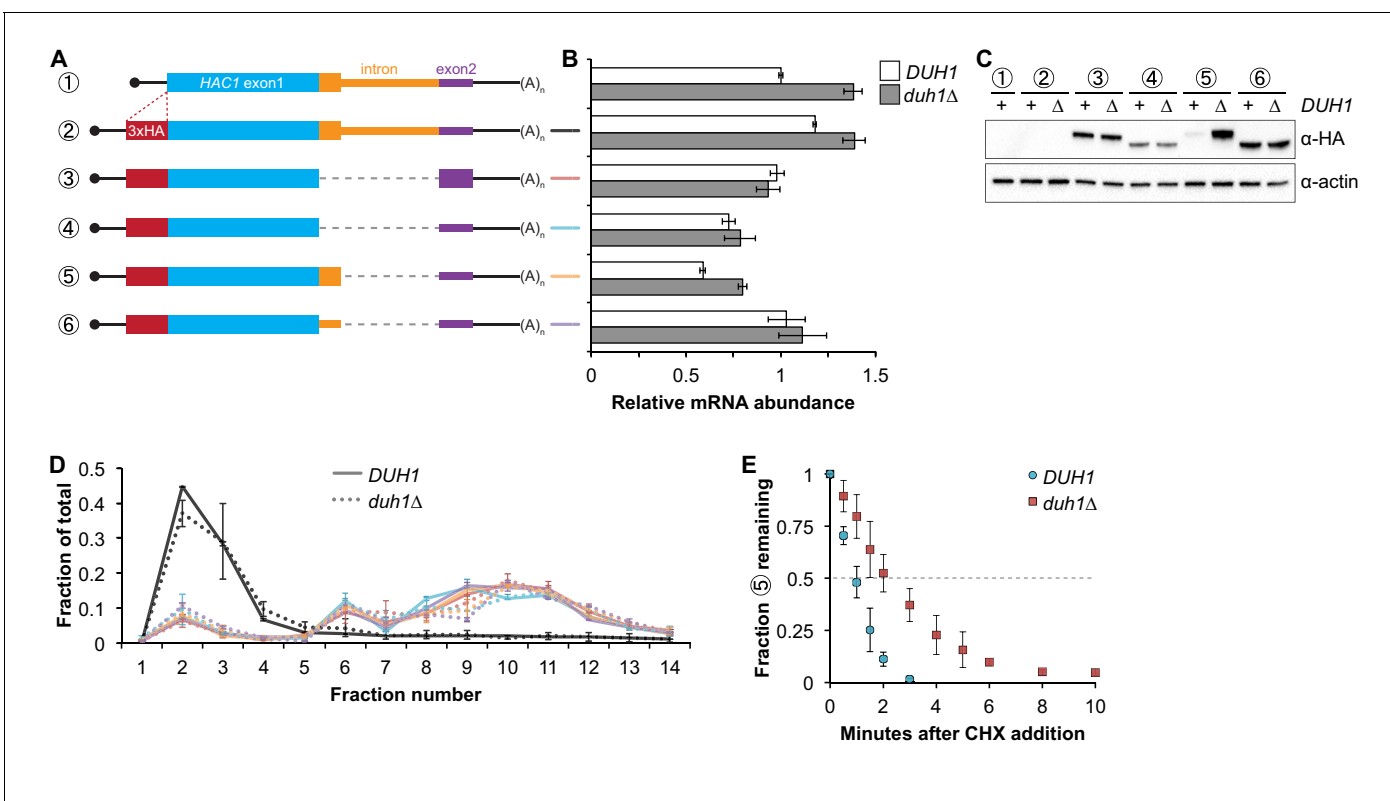

**Figure 5.** Effects of *DUH1* on expression and stability of Hac1p. (**A**) Design of 3xHA-tagged *HAC1* mRNA variants. Constructs are depicted as in *Figure 2B*, with the location of the N-terminal 3xHA tag indicated. (**B**) RNA abundance measurements for *HAC1* mRNA variants. Total RNA was extracted from strains expressing the indicated mRNAs in either a *DUH1* or *duh1Δ* background. qRT-PCR was used to measure the abundances of *HAC1* variants relative to *ACT1* mRNA, with all data normalized to the abundance of construct 1 in strain BY4741. Shown are the mean ± SD (*n* = 2). (**C**) Effect of *DUH1* disruption on protein abundances. Strains expressing the indicated mRNAs depicted in (**A**) in either a *DUH1* or *duh1Δ* background were grown to mid-log phase. Extracts were prepared and immunoblotted for 3xHA-Hac1p and actin loading control. (**D**) Polysome analysis of 3xHA-tagged *HAC1* mRNA variants. Extracts were prepared in heparin-containing lysis buffer from strains expressing the mRNAs indicated in (**A**) in either a *DUH1* or *duh1Δ* background. Polysome analysis was performed as in *Figure 1A*. (**E**) Analysis of protein degradation kinetics. Strains expressing construct 5 (depicted in **A**) in either a *DUH1* or *duh1Δ* background were grown to mid-log phase before being treated with cycloheximide (CHX) to halt translation. At the indicated time points, aliquots of cells were quenched in dry-ice-cold methanol and harvested by centrifugation. Protein extraction and immunoblotting were performed as in (**C**), except that a high-sensitivity antibody was used to detect 3xHA-Hac1^up. Shown are the mean ± SD (*n* = 3), expressed as a fraction of protein detected at *t* = 0.

are regulated by *DUH1*, we constructed a set of HA-tagged strains that constitutively produced either Hac1$^i$p containing the 18-amino-acid exon 2–encoded tail (construct 3), Hac1$^u$p containing the 10-amino-acid intron-encoded tail (construct 5), or Hac1$^{\Delta tail}$p containing no tail at all (constructs 4 and 6). All of the mRNAs encoding HA-tagged Hac1p variants were expressed at similar levels in the presence versus the absence of *DUH1* (*Figure 5B*). Strikingly, at the protein level only the abundance of Hac1$^u$p was affected by disruption of *DUH1*, increasing by ~five fold in the knock-out strain (*Figure 5C*). The increased abundance of Hac1$^u$p in the absence of *DUH1* was not due to increased translation, as evidenced by deletion of *DUH1* having no impact on ribosome density on any of the HA-tagged reporter mRNAs (*Figure 5D*). Instead, our results suggest that Duh1p specifically affects the turnover of Hac1$^u$p due to its 10-amino-acid tail, as was suggested by the results of our reporter experiments.

Because we were able to detect Hac1$^u$p by immunoblotting (using a high-sensitivity antibody) even when *DUH1* was intact (unlike the corresponding GFP reporter), we could use cycloheximide (CHX) shut-off experiments to directly assay the impact of *DUH1* on the turnover of Hac1$^u$p. In both the presence and absence of *DUH1*, Hac1$^u$p was degraded so rapidly that we could not accurately measure its half life even using a rapid harvesting procedure, due to the ~2 minutes required for CHX to accumulate in cells and halt translation (*Gerashchenko and Gladyshev, 2014*). However, the protein-degradation kinetics allowed us to calculate an upper bound for the half life of Hac1$^u$p, which was 50 seconds when *DUH1* was present (*Figure 5E*). Deletion of *DUH1* stabilized Hac1$^u$p and increased its half-life upper bound to 2 minutes. These results demonstrate that *DUH1* is required for the extremely short half life of Hac1$^u$p that normally limits its accumulation. The true half-life difference upon *DUH1* deletion is likely to be greater than the ~2-fold difference in upper bounds that we measured, based on the ~5-fold difference in steady-state protein levels that could not be accounted for by differences in either mRNA abundance or ribosome density (*Figure 5B and D*).

## Synergy between long-range base pairing and Duh1p-dependent degradation

It was previously observed that disrupting the base-pairing interaction between the 5′-UTR and intron of *HAC1* mRNA was sufficient to allow accumulation of Hac1$^u$p, which led to a model in which base-pairing alone was responsible for the post-transcriptional silencing phenomenon (*Rüegsegger et al., 2001*). Our results using constructs in which the base-pairing region was deleted (*Figure 5*) suggested that the previously observed accumulation of Hac1$^u$p was unknowingly being buffered by Duh1p-dependent degradation. To directly address this possibility, we generated HA-tagged constructs in which the base-pairing region was disrupted by mutations in either the 5′-UTR or intron or was reconstituted by the compensatory mutations (*Figure 6A*) and determined the effect of *DUH1* deletion on steady-state protein levels. Because the 5′ and 3′ splice sites remained intact in these constructs, we introduced them into an *ire1Δ* background to eliminate any potential confounding effects of background splicing (*Figure 1—figure supplement 1B*). As previously observed, mutating the base-pairing region in the presence of *DUH1* resulted in detectable levels of Hac1$^u$p (*Figure 6C*). However, the accumulation of Hac1$^u$p was greatly stimulated by deletion of *DUH1*, which was not explained by a corresponding increase in mRNA abundance (*Figure 6B and C*). Thus, although disrupting the base pairing produces detectable amounts of Hac1$^u$p as previously reported (*Rüegsegger et al., 2001*), Duh1p-dependent degradation restricts the steady-state level of the protein.

Remarkably, even a single nucleotide change in the center of the base-pairing region (*Sathe et al., 2015*) was sufficient for some accumulation of Hac1$^u$p, which again was enhanced by deletion of *DUH1* (*Figure 6D*). This result suggests that the 5′-UTR–intron base-pairing interaction is only marginally stable, which may be required for efficient dissociation of the intron after splicing (see Discussion).

## Functional consequences of incomplete silencing

Collectively, we have shown that a pair of silencing mechanisms, one translational and the other post-translational, prevents spurious production of Hac1$^u$p in the absence of the UPR. The existence of such a fail-safe silencing mechanism implies that ectopic production of Hac1$^u$p has physiological

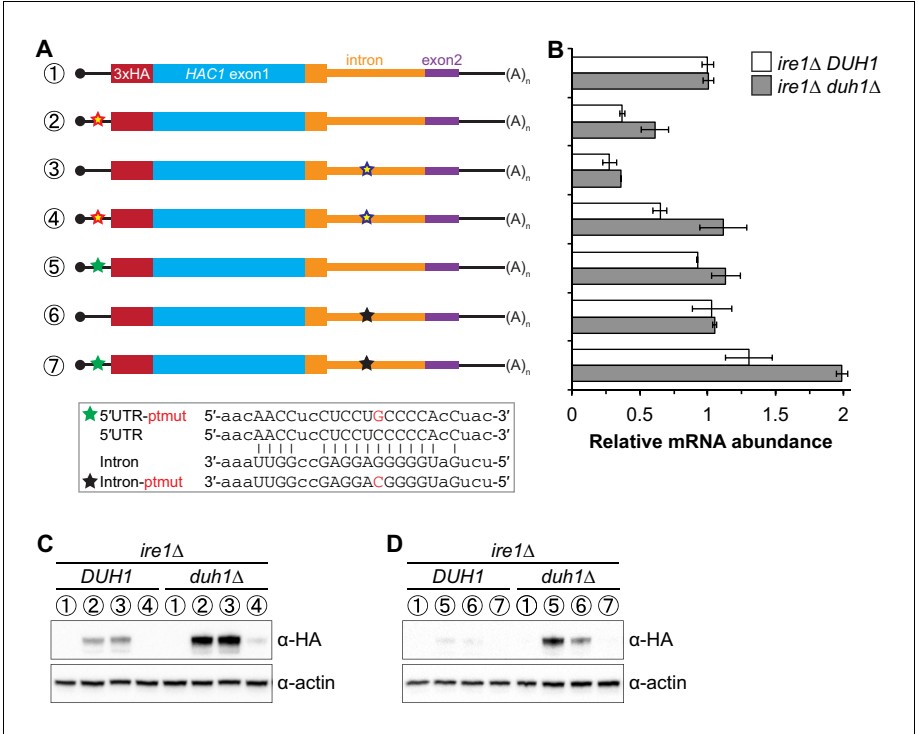

**Figure 6.** Relationship between base pairing– and degron-dependent repression. (**A**) Design of 3xHA-tagged *HAC1* mRNA variants. Constructs are depicted as in *Figure 2B*. Colored stars indicate mutations to the base-pairing region, with specific nucleotide changes shown in *Figure 2B* (red and blue stars) or below (green and black stars) in red. (**B**) RNA abundance measurements for *HAC1* mRNA variants in the indicated strain backgrounds, analyzed as in *Figure 5B*. (**C–D**) Effect of *DUH1* disruption on protein abundances. *ire1Δ* strains expressing the indicated mRNAs depicted in (**A**) in either a *DUH1* or *duh1Δ* background were analyzed as in *Figure 5C*, except that a high-sensitivity antibody was used to detect 3xHA-Hac1$^u$p.

consequences that negatively impact cellular fitness. However, a previous study suggested that Hac1$^u$p is itself not an active transcription factor because it lacks the activating 18-amino-acid tail found in Hac1$^i$p (*Mori et al., 2000*), raising the question as to why Hac1$^u$p accumulation would need to be tightly regulated. We hypothesized that Hac1$^u$p was in fact an active transcription factor but that in the previous study its accumulation had been prevented due to the experiments being performed in a *DUH1* background. To test this hypothesis, we evaluated the ability of strains that constitutively produced either Hac1$^i$p, Hac1$^u$p, or Hac1$^{\Delta tail}$p to grow under conditions of chronic ER stress induced by the drug tunicamycin. Strains expressing Hac1$^i$p or Hac1$^{\Delta tail}$p grew on tunicamycin-containing medium regardless of whether *DUH1* (or *IRE1*) was present (*Figure 7A*), consistent with both proteins being active transcription factors that are not targeted by Duh1p. In contrast, strains expressing Hac1$^u$p only grew robustly on tunicamycin-containing medium when *DUH1* was knocked out (but independently of *IRE1*). These results confirm our hypothesis that Duh1p-dependent degradation normally masks the activity of Hac1$^u$p. Our findings also explain how *HAC1* was able to be initially identified as a high-copy activator of the UPR in an *Δire1* strain, since the UPR activity detected in this strain had to have resulted from Hac1$^u$p produced from unspliced *HAC1* mRNA (*Chapman and Walter, 1997*; *Cox and Walter, 1996*).

Because Hac1$^u$p has UPR-inducing activity (*Figure 7A*), we reasoned that the fail-safe mechanism we discovered is required to prevent leaky production of Hac1$^u$p that would otherwise cause Ire1p-independent activation of the UPR. However, we initially failed to detect Hac1$^u$p produced from unspliced *HAC1* mRNA even when *DUH1* was disrupted (*Figure 5C*). On the other hand, our results from the analogous *GFP* and *HIS3* reporter gene studies demonstrated that translational repression mediated by 5'-UTR–intron base pairing was incomplete and allowed a low level of protein synthesis that was normally 'cleaned up' by Duh1p-dependent degradation (*Figure 4D and E*). This led us to

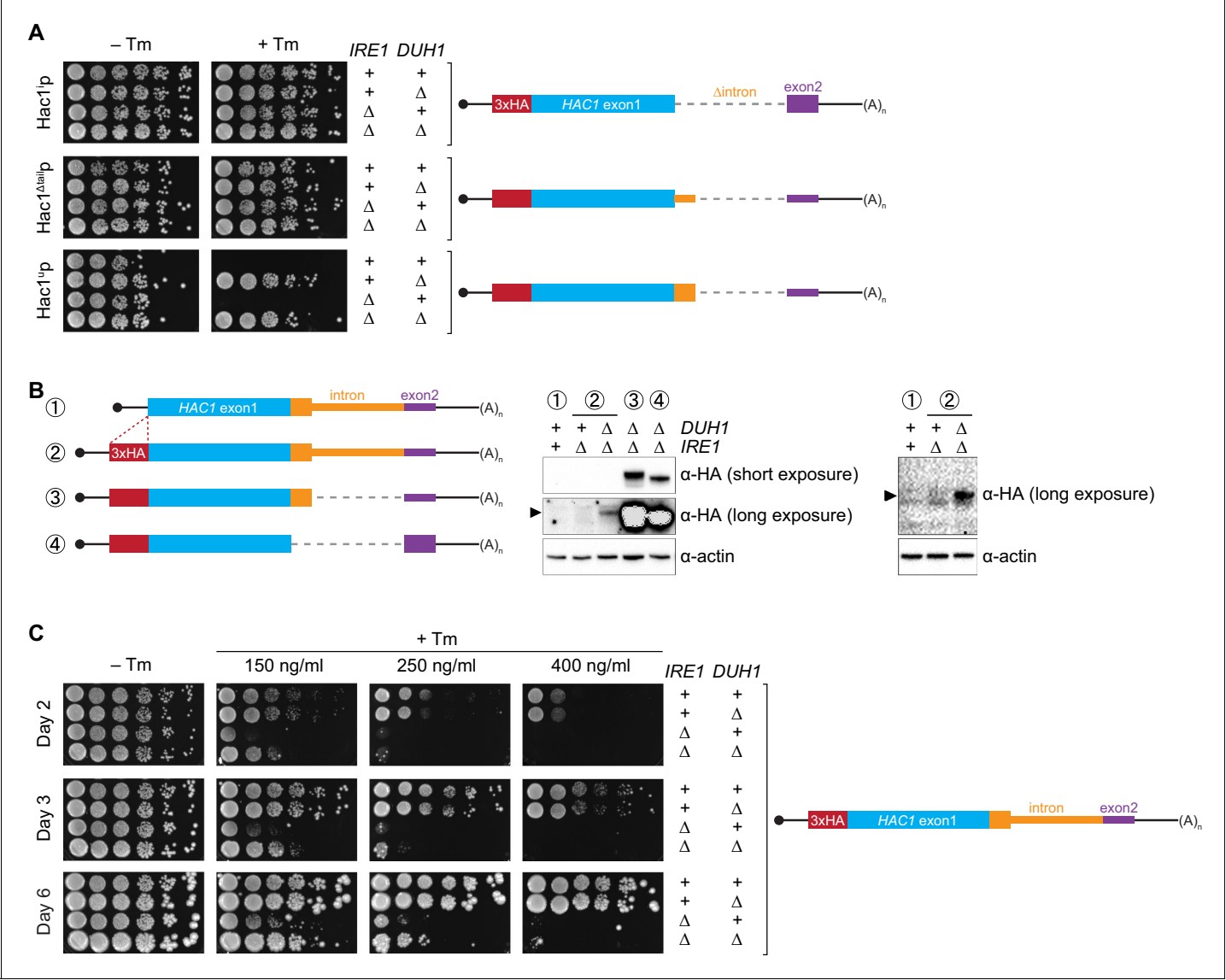

**Figure 7.** Requirement for *DUH1* to suppress Ire1p-independent activation of the UPR. (**A**) Analysis of Hac1p activity in the UPR. Strains expressing the indicated *HAC1* mRNA variants, with (+) or without (Δ) *IRE1* and/or *DUH1* present, were grown to saturation. 10-fold dilution series were plated on YPD without (−Tm) or with (+Tm) 400 ng/ml tunicamycin to induce ER stress. (**B**) Impact of *DUH1* on detection of Hac1$^u$p. Strains expressing the indicated mRNAs, with (+) or without (Δ) *IRE1* and/or *DUH1* present, were analyzed as in *Figure 6C*. Black arrow indicates the position of Hac1$^u$p, which migrates more slowly than Hac1$^i$p. Construct 1, which lacks a 3xHA tag, was used as a negative control for anti-HA immunoblotting. (**C**) Effect of Duh1p-dependent degradation on the UPR. Strains expressing wild-type *HAC1* with an N-terminal 3xHA tag, with (+) or without (Δ) *IRE1* and/or *DUH1* present, were grown to saturation. 10-fold dilution series were plated on YPD without tunimacyin (−Tm) or containing the indicated concentration of tunicamycin (+Tm). Plates were imaged at days 2 (top), 3 (middle), and 6 (bottom).

The following figure supplement is available for figure 7:

**Figure supplement 1.** Ire1p-independent accumulation of Hac1p.

hypothesize that Hac1$^u$p itself was also being occasionally produced from *HAC1$^u$* mRNA and rapidly degraded in a Duh1p-dependent manner, but that the short half life of Hac1$^u$p (*Figure 5E*)—relative to both GFP (*Natarajan et al., 1998*) and His3p (*Belle et al., 2006*)—further reduced the steady-state abundance of Hac1$^u$p to an extremely low level that was initially undetectable.

To enhance detection of HA-tagged Hac1$^u$p, we made two modifications: We used a more sensitive anti-HA antibody for immunoblotting; and we knocked out *IRE1* in our strains, which eliminates the background Ire1p-dependent splicing of *HAC1* mRNA (*Figure 1—figure supplement 1B*) and concomitant production of Hac1$^i$p that can otherwise dominate the signal on immunoblots. With these modifications we could now detect Hac1$^u$p being produced from unspliced *HAC1* mRNA, but only when the protein was stabilized by deletion of *DUH1* (*Figure 7B*) and at levels far below even those of Hac1$^u$p being constitutively produced in the presence of *DUH1* (*Figure 7—figure supplement 1*). Despite the relatively low level of leaky translation product we detected, these results provide molecular evidence that Hac1$^u$p is being continuously produced from *HAC1$^u$* mRNA but rapidly degraded due to its C-terminal degron.

Does the leaky production of Hac1$^u$p unmasked by deletion of *DUH1* have any functional consequences? The ability of constitutively produced Hac1$^u$p to promote survival under UPR-inducing conditions (*Figure 7A*) suggested that small amounts of Hac1$^u$p might also induce the UPR to some extent. To test this possibility, we examined how strains expressing HA-tagged but otherwise unmodified *HAC1* mRNA grew on media containing different concentrations of tunicamycin. At the highest tunicamycin concentration only strains expressing *IRE1* could grow regardless of whether *DUH1* was also present (*Figure 7C*), suggesting that growth was dependent on the abundant Hac1$^i$p produced from *HAC1$^i$* mRNA. In contrast, at lower concentrations of tunicamycin the *IRE1* deletion strain showed some growth, but this was reproducibly enhanced by simultaneous deletion of *DUH1*. These results indicate that the low level of Hac1$^u$p detectable in strains lacking *DUH1* (*Figure 7B*) is sufficient to activate the UPR enough to facilitate cell survival under stress, even in the absence of Ire1p. Thus, degron-dependent degradation of Hac1$^u$p mediated by Duh1p is normally required to prevent ectopic Ire1p-independent activation of the UPR.

## Discussion

Altogether, we have shown that the ability of unspliced *HAC1* mRNA to be stored in the cytoplasm without giving rise to detectable protein depends on two layers of post-transcriptional silencing, which together comprise a fail-safe mechanism (*Figure 8*). The initial silencing mechanism that acts on *HAC1$^u$* mRNA is a block to translation initiation, which is caused by secondary structure in the 5′-UTR that inhibits binding or progression of the scanning ribosome. However, this silencing mechanism does not entirely prevent translation, allowing the production of small amounts of Hac1$^u$p. Because Hac1$^u$p is an active transcription factor, even small amounts of this protein produced by leaky translation could activate the UPR and affect cellular physiology. To prevent this, a second silencing mechanism exists to rapidly degrade any Hac1$^u$p that is produced by leaky translation. By relying on recognition of the unique C-terminal tail of Hac1$^u$p by the F-box protein Duh1p, the protein-degradation mechanism selectively targets Hac1$^u$p for ubiquitination and subsequent degradation by the proteasome. Because both translational repression and protein degradation rely on sequences found in the intron of *HAC1*, removal of the intron by Ire1p-dependent splicing simultaneously results in the inactivation of both silencing mechanisms. In this way, splicing allows the rapid accumulation of Hac1$^i$p from the existing pool of *HAC1* mRNA.

One surprising finding of our studies is that the polysome-like sedimentation of *HAC1$^u$* mRNA is not due to translation of the mRNA. Instead, our results using mRNAs containing premature stop codons and using a modified lysis buffer (*Figure 1*) indicate that translation initiation on *HAC1$^u$* mRNA almost never occurs due to the 5′-UTR–intron base-pairing interaction. The lack of appreciable translation of *HAC1$^u$* mRNA suggests that the substrate for Ire1p-dependent splicing is untranslated mRNA, rather than polysomal mRNA containing stalled ribosomes as originally proposed (*Rüegsegger et al., 2001*). A corollary of this is that the synthesis of Hac1$^i$p requires *de novo* translation initiation on *HAC1$^i$* mRNA, rather than just the resumption of translation by stalled ribosomes that initiated on *HAC1$^u$* mRNA. The initial hint that *HAC1$^u$* mRNA is translationally repressed at the initiation rather than elongation stage came from ribosome-profiling data, which revealed a lack of ribosome-protected fragments derived from *HAC1$^u$* mRNA (*Weinberg et al., 2016*). At least in this case, our findings suggest that ribosome profiling can provide a more accurate measurement of translation than traditional polysome profiling.

What is the molecular basis for the polysome-like sedimentation of *HAC1$^u$* mRNA? The disruption of the pseudo-polysomes by addition of heparin, which is more negatively charged than RNA itself,

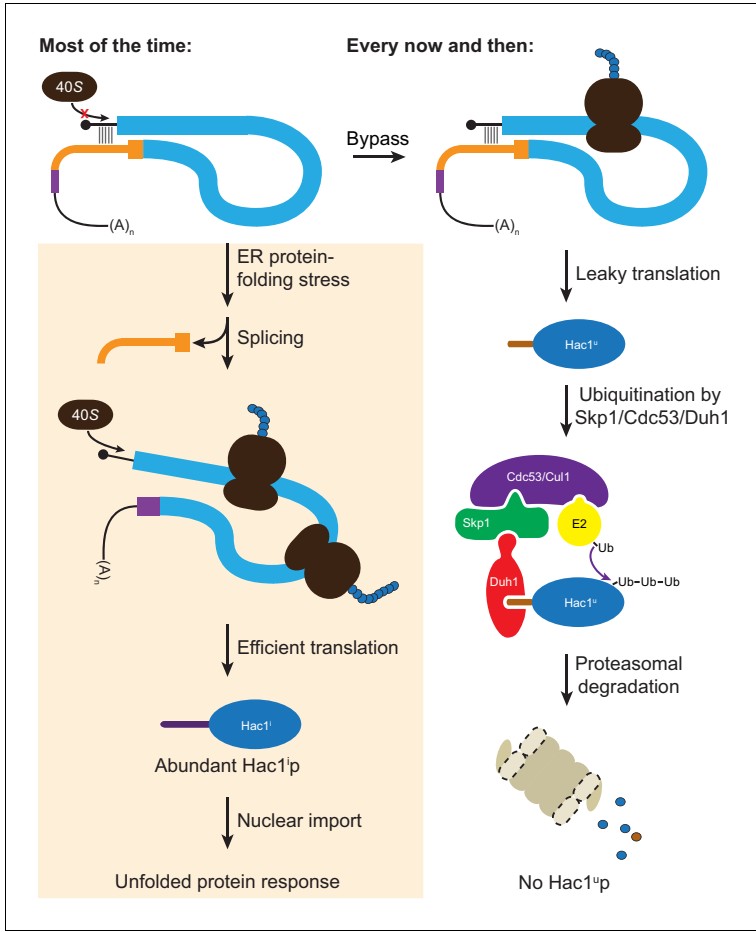

**Figure 8.** Fail-safe post-transcriptional silencing of unspliced *HAC1* mRNA. See the main text for a description.

suggests that electrostatic interactions are responsible. Moreover, the discontinuous distribution of $HAC1^u$ mRNA that matches the peaks and valleys observed for polysomes (*Rüegsegger et al., 2001*) indicates that $HAC1^u$ mRNA is associated with integral numbers of actual 80$S$ ribosomes. On the basis of these data, we propose that electrostatic interactions between $HAC1^u$ mRNA and either non-translating 80$S$ ribosomes or, more likely, bona fide polysomes (containing both ribosomes and mRNA) are responsible for its polysome-like sedimentation. $HAC1^u$ mRNA may be especially prone to such non-specific interactions because of the strong initiation block, though other untranslated mRNA molecules (including the small fraction of ribosome-free mRNA molecules observed even for well-translated genes) may also behave similarly. In this regard, the analysis method (i.e., use of constructs containing premature stop codons) and experimental tools (i.e., exogenous RNA controls combined with heparin-containing lysis buffer) that we utilized here to definitively establish the translation state of $HAC1^u$ mRNA will be generally useful in polysome-profiling studies in the future.

After the unexpected discovery that eliminating the 5′-UTR–intron base pairing alone did not result in detectable protein expression, we uncovered the effect of protein degradation caused by an intron-encoded C-terminal degron. The genetic selection that we performed to identify components of the protein-degradation pathway yielded just a single gene with recurrent mutations: *DUH1*. Our failure to isolate mutations in other components of the ubiquitin–proteasome machinery (i.e., the E2 ubiquitin-conjugating enzyme that presents the ubiquitin to the Duh1p-containing SCF complex for transfer onto $Hac1^u$p; and subunits of the proteasome itself) may reflect the pleiotropic nature of such mutations, which would hinder growth despite the *HIS3* reporter gene being de-silenced; an extreme case is the essential gene encoding the sole E1 ubiquitin-activating enzyme in yeast, *UBA1*. In addition, our reliance on spontaneous mutagenesis made it unlikely that we would

have isolated multiple mutations in redundant sets of genes, which may also explain our failure to isolate E2 mutants given their sometimes overlapping functions (*Chen et al., 1993*). Nonetheless, the fact that we isolated 16 different mutations in *DUH1* in 17 of our 20 sequenced strains illustrates the power of our genetic selection system. A noteworthy feature of our selection was that by combining spontaneous mutagenesis with whole-genome sequencing, we were able to rapidly go directly from phenotypic mutants to genotypic mutations without any crosses and relying on only simple computational tools for analysis.

Compared to an alternative silencing mechanism involving only protein degradation with no translational repression (*Cox and Walter, 1996*), the fail-safe mechanism we discovered has the advantage of not wasting resources on the production of Hac1$^u$p that will anyway be rapidly degraded. But why does budding yeast rely on a pair of silencing mechanisms to prevent accumulation of Hac1$^u$p, rather than just completely blocking translation in the first place? An intriguing possibility is that the marginal stability of the 5′-UTR–intron base-pairing interaction (*Figure 6D*) is the cause of leaky translation but is also required to make the translational repression reversible (*Rüegsegger et al., 2001*). If the base-pairing interaction were more stable and prevented translation altogether, splicing might not be sufficient for the intron to dissociate from the 5′-UTR despite their being covalently unlinked. Another possibility is that the act of translating the mRNA has a function. For example, a pioneer round of translation on *HAC1$^u$* mRNA might be required to form the 5′-UTR–intron interaction that inhibits subsequent rounds of translation. Such a requirement might be due to the ribosome physically bringing the distant sequences together through a tethering mechanism, or due to the ribosome unwinding alternative competing structures that initially form with the ORF. A third possibility is that the small amount of Hac1$^u$p being constantly produced and degraded has a function. In particular, the ability to regulate accumulation of Hac1$^u$p through Duh1p without relying on Ire1p activation would provide a means of rapidly producing a small pool of Hac1$^u$p in the absence of ER protein-folding stress. A constitutively produced yet transient pool of Hac1$^u$p might also contribute to the switch-like behavior of the UPR, based on the predicted ability of Hac1$^u$p to heterodimerize with Hac1$^i$p and thereby facilitate proteasomal degradation of any Hac1$^i$p produced from background splicing (*Figure 1—figure supplement 1*).

Alternatively, the leaky translation of *HAC1$^u$* mRNA that we discovered may reflect a more fundamental property of translational repression: that it is inherently incomplete. The two predominant mechanisms known to inhibit cap-dependent translation are upstream ORFs (uORFs) and 5′-UTR secondary structure (*Gebauer and Hentze, 2004*). Inhibition by uORFs has been shown to be only partial, due to both leaky scanning enabling bypass of the uORF and reinitiation facilitating translation of the downstream ORF (*Meijer and Thomas, 2002*). Our findings suggest that inhibition by 5′-UTR secondary structure may also be incomplete, potentially due to the action of RNA helicases that are constantly unwinding RNA structures *in vivo* (*Rouskin et al., 2014*). Future work may shed light on both the causes and consequences of the incomplete translational repression of unspliced *HAC1* mRNA.

## Materials and methods

### Yeast growth

*Saccharomyces cerevisiae* strains used in this study were derived from BY4741 (*MATa his3Δ1 leu2Δ0 met15Δ0 ura3Δ0*) and are listed in *Supplementary file 1*. Strains were cultured at 30°C with shaking in rich yeast-peptone-dextrose (YPD) medium, synthetic complete (SC) medium, or the appropriate drop out medium (e.g., SC–His) as indicated. Cultures were grown to mid-log phase (OD$_{600}$ ~0.5) before harvesting by either vacuum filtration (polysome analysis), methanol quenching (protein half-life), or brief centrifugation at 4°C (all other experiments) and then snap frozen in liquid nitrogen. For growth tests under UPR stress (*Figure 7*), 10-fold serial dilutions of cells were spotted onto YPD plates containing either the indicated concentration of tunicamycin (Sigma, St. Louis, MO) dissolved in DMSO (+Tm) or DMSO alone (–Tm).

### Yeast strain construction

To facilitate generation of *HAC1* variant and reporter strains, the entire transcribed region of the *HAC1* gene – as well as additional flanking sequence – was amplified from genomic DNA (extracted

from BY4741) and cloned into the pCR4-TOPO vector backbone (Invitrogen, Carlsbad, CA) using Gibson Assembly Master Mix (New England Biolabs, Ipswich, MA) according to the Gibson assembly method (*Gibson et al., 2009*). *GFP* and *HIS3* reporter constructs were generated from this plasmid by replacing the *HAC1* ORF with the reporter gene using the Gibson assembly method. Variants of these constructs were generated using inverse PCR with 5′-phosphorylated primers (designed to introduce point mutations or deletions) and Phusion High-Fidelity DNA Polymerase (New England Biolabs) followed by ligation with Quick Ligase (New England Biolabs); or using the Gibson assembly method with either PCR products or synthetic gBlocks (Integrated DNA Technologies, Coralville, IA). All plasmid inserts were verified by Sanger sequencing before PCR amplification with Phusion DNA polymerase for integration into the yeast genome.

Yeast transformations were performed using the PEG–lithium acetate method (*Gietz and Woods, 2002*). *HAC1* was deleted from the genome using the *URA3* cassette of pRS416 to replace the entire transcribed region of *HAC1* (from 366 nucleotides upstream of the start codon, to 750 nucleotides downstream of the stop codon in exon 2). *IRE1* and/or *DUH1* were deleted using the *HIS3MX* or *URA3* cassettes of pFA6a-3HA-HIS3MX6 or pRS416, respectively, as indicated in *Supplementary file 1*. *HAC1* variant and reporter strains were generated from *hac1Δ::URA3* strains using 5-FOA counterselection. Transformants were screened by PCR to identify integrants, which were subsequently verified by PCR and Sanger sequencing of the entire integrated cassette.

## Sucrose gradient analysis

Polysome analyses by sucrose gradient fractionation are described in more detail at Bio-protocol (*Aboulhouda et al., 2017*). Yeast strains were grown to mid-log phase ($OD_{600} \sim 0.5$) in YPD and harvested by vacuum filtration as previously described (*Weinberg et al., 2016*). The frozen cell pellet was transferred into a pre-chilled mortar that was surrounded by and filled with liquid nitrogen. The pellet was ground to a fine powder by hand using a pre-chilled pestle, transferred into a 50 ml conical tube filled with liquid nitrogen, and after boiling off the liquid stored at –80°C. Crude lysate was prepared by briefly thawing the cell powder on ice for 1 min and then resuspending in 1 ml polysome lysis buffer (20 mM HEPES-KOH [pH 7.4], 5 mM $MgCl_2$, 100 mM KCl, 1% Triton X-100, 2 mM DTT, 100 µg/mL cycloheximide, 20 U/mL SuperUPERase-In [Thermo Fisher Scientific, Waltham, MA], 1X cOmplete EDTA-free Protease Inhibitor Cocktail [Roche, Switzerland]). Where indicated, the lysis buffer also contained 200 µg/mL heparin (Sigma). The lysate was centrifuged at 1300 *g* for 10 min, and the supernatant was transferred to a new tube taking care to avoid the thick top lipid layer. The resulting clarified lysate was diluted to a concentration of 25 $OD_{260}$ units/ml, flash frozen in liquid nitrogen in single-use aliquots, and stored at –80°C.

A 10–50% continuous sucrose gradient was prepared in polysome gradient buffer (20 mM HEPES-KOH [pH 7.4], 5 mM $MgCl_2$, 100 mM KCl, 2 mM DTT, 100 µg/mL cycloheximide, 20 U/mL SUPERase-I) and allowed to cool to 4°C while lysate was thawed gently on ice. Where indicated, 100 ng luciferase RNA (Promega, Madison, WI) was then added to 100 µl of thawed cell lysate. 100 µl of cell lysate (at 25 $OD_{260}$ units/ml) was loaded on top of the gradient, and gradients were spun in an SW41-Ti rotor at 36000 rpm for 2.5 hr at 4°C. Gradients were fractionated into 15 fractions using a Biocomp Piston Gradient Fractionator, and fractions were flash frozen and stored at –80°C.

RT-qPCR was performed directly on gradient fractions using the TaqMan Gene Expression Cells-to-CT Kit (Life TechnologiesThermo Fisher Scientific). Briefly, 10 µl Lysis Buffer containing DNase was combined with 0.1 µl synthetic XenoRNA control (provided with the TaqMan Cells-to-CT Control Kit) and 1 µl of the fraction. Reactions were incubated at room temperature for 5 min, and then quenched on ice by addition of 1 µl STOP Solution followed by incubation at room temperature for an additional 2 min. 4.5 µl of this reaction was added to 5.5 µl RT Master Mix, and reverse transcription was carried out at 37°C for 1 hr followed by a 5 min incubation at 95°C before cooling to 4°C. RT reactions were diluted 6-fold in nuclease-free water prior to qPCR.

qPCR was carried out in duplicate for each fraction using the TaqMan Gene Expression Master Mix (Thermo Fisher Scientific) according to the manufacturer's instructions, using 10 µl reactions containing 4.5 µl of six-fold diluted RT reaction per qPCR reaction. XenoRNA was analyzed using the primer-probe assay provided with the TaqMan Cells-to-CT Control Kit. *ACT1* and luciferase RNAs were analyzed using predesigned TaqMan Gene Expression Assays from Life TechnologiesThermo Fisher Scientific (Sc04120488_s1 and Mr03987587_mr, respectively). *HAC1* and

*GFP* RNAs were analyzed using the following primer-probe qPCR assays from Integrated DNA Technologies (containing 6-FAM/ZEN/IBFQ quenchers with a primer-to-probe ratio of 1-to-2):

| Gene | Primer 1 | Primer 2 | Probe |
| --- | --- | --- | --- |
| *HAC1* | TCAAGAGCTATGTTCAGTGTCG | GGTTTCTACTGTTCTGTCTCCG | 56-FAM/CGCGCCCTC/ZEN/CTACAATTATTTGTGG/3IABkFQ |
| *GFP* | GTTTGCCATAAGTTGCGTCC | TGGTAGAATTGGATGGCGAC | 56-FAM/CTGTGAGTG/ZEN/GTGAGGGTGAAGGG/3IABkFQ |

Relative mRNA abundances in each fraction (i.e., Cq values) were first normalized to XenoRNA to account for differences in qRT-PCR efficiency among fractions, and then calculated as a percentage of total mRNA detected across the gradient. All strains were analyzed in duplicate, beginning with separate cultures of the same strain, to confirm the reproducibility of each result. *ACT1* mRNA was analyzed in every gradient and confirmed to co-sediment with polysomes in all cases.

## Flow cytometry analysis

Yeast strains were grown to mid-log phase ($OD_{600} \sim 0.5$) in SC medium, then diluted 10 fold in fresh medium before quantifying GFP fluorescence using a LSR II flow cytometer (Becton Dickinson, Franklin Lakes, NJ) and 530/30 filter. Raw data from 10,000 cells per sample was analyzed using FlowJo and gated to remove debris before being exported to Microsoft Excel. Fluorescence in each cell was normalized to cell size using the side scatter measurement before calculating the median and quartiles of the population. All strains were analyzed at least twice, beginning with separate cultures of the same strain, to confirm reproducibility. The equivalent of 5 $OD_{600}$ units was harvested for both protein and RNA analyses from the same cultures used for flow cytometry.

## Protein analysis

Protein samples were prepared using the NaOH/TCA method (*Riezman et al., 1983*), separated by SDS-PAGE using 12% or 4–12% Bolt Bis-Tris gels (Thermo Fisher Scientific), and transferred in 1X CAPS Buffer onto 0.22 micron PVDF membrane (Bio-Rad, Hercules, CA). Blots were probed with the following antibodies diluted in 1X TBS-T containing 5% nonfat dry milk: mouse anti-GFP (RRID:AB_390913, Roche #11814460001, 1:1000), mouse anti-HA (RRID:AB_627809, Santa Cruz Biotechnology [Dallas, TX] sc-7392, 1:3000), rat anti-HA high sensitivity (RRID:AB_390918, Roche #11867423001, 1:5000), mouse anti–beta actin (RRID:AB_449644, AbCam [] ab8224, 1:10000), HRP-conjugated goat anti-mouse IgG (RRID:AB_631736, Santa Cruz Biotechnology sc-2005, 1:10000), and HRP-conjugated goat anti-rat IgG (RRID:AB_631755, Santa Cruz Biotechnology sc-2032, 1:10000). Blots were developed using Clarity ECL Western Blotting Substrate (Bio-Rad), and chemiluminescence was detected on a ChemiDoc Imaging System (Bio-Rad).

## RNA analysis

Total RNA was isolated using the hot acid phenol method (*Ares, 2012*) and resuspended in nuclease-free water. For each sample, 2.5 µg of RNA was treated with RQ1 DNase (Promega) according to the manufacturer's instructions. cDNA synthesis and qPCR were performed using the Cells-to-CT Kit (Thermo Fisher Scientific) as described above.

For RT-PCR analysis of *HAC1* splicing, cDNA generated from input samples (as described in 'Sucrose gradient analysis') was used as a template for PCR amplification using the following intron-flanking primers: forward, ACGACGCTTTTGTTGCTTCT; reverse, TCTTCGGTTGAAGTAGCACAC. PCR products were analyzed by agarose gel electrophoresis.

## Protein half-life determination

Yeast strains YRDS221 and YRDS241 (*Supplementary file 1*) were grown to mid-log phase ($OD_{600} \sim 0.5$) in SC medium. Cycloheximide (Sigma) was added to a final concentration of 100 µg/ml and samples collected at the indicated time points. To rapidly harvest samples, 5 ml of culture was directly added to a prechilled 50 ml tube (in a dry ice–ethanol bath) containing 25 ml Quench Solution (60% methanol, 10 mM HEPES-KOH pH 7.4) and mixed well. Cells were collected by centrifugation at 4°C, snap frozen, and processed for immunoblotting as above. Quantification of immunoblot

signal was performed by densitometry with ImageJ. An upper bound on protein half-life was calculated from 3 independent experiments as the time point at which ~50% of the protein remained.

## Genetic selection and whole-genome sequencing

Yeast strain YRDS57 (*Supplementary file 1*) was grown overnight in liquid YPD, plated on SC–His solid medium, and allowed to grow for 5–7 days at 30°C. Spontaneous mutant colonies were isolated from five independent platings and verified for growth on SC–His by restreaking and for expression of the HA-His3p by immunoblotting. Clones were then examined for *cis* mutations by Sanger sequencing the 3′ end of the *HIS3* reporter gene. Of the remaining clones that had a 'trans' mutation, 20 were chosen for analysis by genomic DNA sequencing along with 4 control strains (the parent strain YRDS57 and 3 strains with *cis* mutations).

Genomic DNA was isolated from individual saturated overnight cultures using the MasterPure Yeast DNA Purification Kit (Epicentre) according to the manufacturer's instructions. Purified genomic DNA was treated with RNase A/T1 Cocktail (Thermo Fisher Scientific), phenol extracted, precipitated with ethanol, and quantified with the Qubit dsDNA HS Assay Kit (Thermo Fisher Scientific). For each strain 1 ng of genomic DNA was used as input into the Nextera XT DNA Library Preparation Kit (Illumina, San Diego, CA), and genomic DNA libraries were prepared according to the manufacturer's instructions using the Nextera XT Index Kit for barcoding during PCR. Genomic DNA libraries from each strain were quantified with the Qubit dsDNA HS Assay Kit (Invitrogen), and pooled into a single tube in equal proportions. Pooled DNA was purified using the DNA Clean and Concentrator-5 Kit (Zymo Research, Irvine, CA), and 200–350 bp fragments were isolated using a BluePippin Gel Cassette (2% agarose dye-free with internal standards; ). The size-selected library was sequenced on a single lane of a HiSeq 4000 sequencer using 50-nucleotide single-end reads.

Data analysis was performed using the Galaxy web server (*Afgan et al., 2016* ). Reads in FASTQ format were aligned to the sacCer3 reference genome using BWA (version 0.1) with default 'Commonly Used' settings. Sequence differences between the reference genome and aligned reads were identified using FreeBayes (version 0.4) to generate a variant call format (VCF) file for each strain. The VCF-VCFintersect tool (with options '-i –invert') was then used to compare VCF files for mutants against the VCF file for the parental selection strain, in order to identify variants that were unique to the mutants. Finally, variants were manually annotated using the UCSC Genome Browser. For proof-of-principle analysis of *cis* mutants, the same mapping and variant-calling procedures were used with a custom reference genome containing only the reporter gene.

## Accession numbers

Genome-sequencing data from the genetic selection have been deposited in the NCBI Sequence Read Archive (SRA) under accession SRP081128.

## Acknowledgements

We thank Peter Walter, David Toczyski, Graeme Gowans, and members of the Weinberg lab for insightful discussions and comments on the manuscript, Manuel Leonetti and Meghan Zubradt for assistance with flow cytometry, and Eric Chow for advice about high-throughput sequencing. We also acknowledge Jonathan Weissman, Raul Andino, Keith Yamamoto, and Alan Frankel for generously sharing equipment. High-throughput sequencing was performed at the UCSF Center for Advanced Technology. This work was supported by the UCSF Program for Breakthrough Biomedical Research (funded in part by the Sandler Foundation) and by an NIH Director's Early Independence Award (DP5OD017895).

## Additional information

### Funding

| Funder | Grant reference number | Author |
| --- | --- | --- |
| NIH Office of the Director | DP5OD017895 | Rachael Di Santo<br>Soufiane Aboulhouda<br>David E Weinberg |

| University of California, San Francisco | Program for Breakthrough Biomedical Research | Rachael Di Santo Soufiane Aboulhouda David E Weinberg |
|---|---|---|

The funders had no role in study design, data collection and interpretation, or the decision to submit the work for publication.

## Author contributions
RDS, DEW, Conception and design, Acquisition of data, Analysis and interpretation of data, Drafting or revising the article; SA, Acquisition of data, Analysis and interpretation of data

## Author ORCIDs
David E Weinberg, http://orcid.org/0000-0002-9348-1709

# Additional files

## Supplementary files
• Supplementary file 1. Yeast strains used in this study. See key below.

## Major datasets
The following dataset was generated:

| Author(s) | Year | Dataset title | Dataset URL | Database, license, and accessibility information |
|---|---|---|---|---|
| Di Santo R, Weinberg DE | 2016 | Whole-genome sequencing of spontaneous mutants | http://trace.ncbi.nlm.nih.gov/Traces/sra/?study=SRP081128 | Publicly available at the NCBI Sequence Read Archive (accession no: SRP081128) |

The following previously published dataset was used:

| Author(s) | Year | Dataset title | Dataset URL | Database, license, and accessibility information |
|---|---|---|---|---|
| Weinberg DE, Shah P, Eichhorn SW, Hussmann JA, Plotkin JB, Bartel DP | 2016 | Data from: Improved Ribosome-Footprint and mRNA Measurements Provide Insights into Dynamics and Regulation of Yeast Translation | http://www.ncbi.nlm.nih.gov/geo/query/acc.cgi?acc=GSE75897 | Publicly available at the NCBI Gene Expression Omnibus (accession no: GSE75897) |

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
