## [Decision Letter]

Thank you for submitting your article "The fail-safe mechanism of post-transcriptional silencing of unspliced *HAC1* mRNA" for consideration by *eLife*. Your article has been favorably evaluated by James Manley (Senior Editor) and three reviewers, one of whom, Rachel Green (Reviewer #1), is a member of our Board of Reviewing Editors. The following individual involved in review of your submission has also agreed to reveal their identity: Traude Beilharz (Reviewer #3).

The reviewers have discussed the reviews with one another and the Reviewing Editor has drafted this decision to help you prepare a revised submission.

Summary:

This is an outstanding manuscript that tells a very complete story about multiple pathways for regulating the expression of Hac1p, such that protein is produced only during conditions of UPR stress. The study includes multiple approaches to address each question including a genetic screen with a novel E3 ligase identified. All three reviewers feel the manuscript is highly suitable for publication in *eLife* with minor changes.

Essential revisions:

The authors should address the citation issues raised by reviewer 2 and the figure issues raised by reviewers 1 and 3 (colors for exons/introns etc.). Finally, some language about the "only" gene that emerged from the strain should be corrected (full reviews included below).

Congratulations on an outstanding study.

*Reviewer #1:*

This is a very complete and compelling manuscript defining the mechanisms though which Hac1 expression is controlled in the absence of UPR stress in *S. cerevisiae*. Contrary to earlier studies, DiSanto et al. establish that Hac1 translation is controlled first at the level of initiation through secondary structural interactions between the 5'UTR and the intron, and secondly, through protein degradation via a novel E3 ligase targeting the intron-encoded 8 aa sequence. The manuscript nicely resolves many small issues in the literature regarding this tightly regulated and biologically critical gene. A few small points are suggested below but this manuscript is highly suitable for publication in *eLife*.

1) The pseudo-polysome phenomenon is nicely addressed experimentally but could be more clearly described in the text. What is observed is A260 that rises about background in a rather amorphous smear – earlier studies that collected more fractions saw the ribosome peaks emerging – this could be discussed more explicitly. To the non-expert, the modest background A260 will not be particularly compelling, and so the modest reduction will not seem compelling either. This is a request for textual changes.

2) In all Figures, the 5' and 3' UTR are colored in black as though they are not part of the exons – for clarity, these should be colored in exon color but in thin lines instead of thick (for coding).

3) RNA level changes are dismissed as not relevant though in some figures (3A for example), there are substantial changes in RNA level that should not be so quickly dismissed.

4) Discussion of why this transcription factor Hac1u needs to be so tightly controlled, but why translational control is not sufficient, was a bit too long and speculative. This portion of the Discussion could be reduced.

*Reviewer #2:*

This manuscript from the Weinberg lab addresses a longstanding mystery in the UPR field: why does the HAC1 mRNA appear in heavy polysomes even though it does not generate any protein product? Existing data supported a model where blockage of initiation by base-pairing between the intron and 5'UTR prevented HAC1 translation. It was assumed that this blockage somehow also stalled 80S elongating ribosomes and thereby accounted for the migration in heavy polysome fractions. Recently, ribosome profiling datasets revealed that the mRNA did not appear to be bound to translating 80S ribosomes. Experiments in this manuscript resolve this contradiction by showing the old polysome profiling methods were flawed and that the base-pairing between intron and 5'UTR does appear to block most translation from initiating. Surprisingly, however, the authors also found that elimination of this base-pairing interaction did not result in protein production, even though it does lead to migration of the mRNA in heavy polysome fractions (as observed by an improved polysome profiling method).

The authors then meticulously build a case for a model where the amino acid sequence encoded by the intron encodes a degron tag. This tag is recognized by a previously unknown E3 ligase that results in degradation of any Hac1 protein that happens to be produced from the unspliced transcript. Presumably, this system is important for eliminating protein produced by low levels of HAC1 translation that escape the initiation-based inhibition mechanism. The paper therefore reveals a new layer of regulation of this regulatory network and opens up many new questions for the field to explore.

The experimental work is very convincing and aside from a few minor points about inadequate citation (see below), I have no major concerns. I think this paper will generate wide interest because it raises so many new questions about UPR regulation and, more generally, the breadth of targets of the ubiquitin proteasome system.

Concern about adequate citation:

The original discovery of Hac1p function by Cox and Walter in 1996 showed that 1) Uninduced Hac1p is an effective stimulator of the UPR and 2) E2 ligases are involved in degrading uninduced Hac1p. Subsequent papers (including most recently work by the Dever and Dey labs) also showed that uninduced Hac1p is capable of conferring resistance to tunicamycin.

Therefore, the authors should eliminate the statement that their results, "confirm our hypothesis that Hac1up is indeed a functional UPR transcription factor." This was already established and should be acknowledged as such. Second, they should point out that the idea of the intron "tail" sequence serving as a degron dates back two decades. While it's true that this model was not strongly embraced by the field, the Walter lab should be cited for first proposing and testing it.

*Reviewer #3:*

I like this manuscript. It seems to me to reconcile multiple conflicts in the literature and to provide new insight in to the regulation of the UPR with a satisfying level of clarity. The data are of high quality and have been clearly designed and executed to probe the precise aspect of the experimental hypotheses in question. E.g., the clever use of reporters/ mutant strains/ and new technologies mean there is no confusion regarding the source of regulatory control. The problem of mRNA co-sedimentation with polysomes in the absence of translation is satisfyingly dealt with, although it will likely to confound the literature for some time to come. The sequencing of spontaneous suppressor mutants to define the second-phase repression by proteolysis is also lovely.

My comments/questions are therefor of a fairly minor nature, but might also be of interest to other readers.

1) In the figures, can the authors unify the schematics for the mRNA structures. Specifically, I thought Figure 1 should have the same colour coding and intron included. At first reading of Figure 1, it took a while to recognise that fat-to-thin line transition meant inclusion of 3' nonsense stop codons. I'm not sure if this needs more explicit marks on the figures. Also should the purple line on the left-hand panel of Figure 8 be fat?

2) On the topic of the truncating stop-codons should there be comment about why these don't trigger NMD? Or is the fact that they don't more evidence toward the point that these mRNA are not translated.

3) At the end of the second paragraph of the subsection “An additional silencing mechanism downstream of translation initiation”: prompted by the point about 5'-3' looping being an impediment to "binding or progress of the scanning ribosome" I checked the recent TCP-seq App from Archer et al. Nature 2016 (http://bioapps.erc.monash.edu/TCP/) to see the HAC1 transcript had any interesting regulation by small subunit scanning. However, there is no-data, which is unexpected given HAC1s abundance. But it does suggest that the inhibition precedes binding of even a single SSU necessary for the TCP-seq approach.

4) In the Discussion: I wondered if an additional purpose or benefit of leaky translation might be that inhibition of proteolysis might be a kinetically quicker way to get a functional transcription factor with which to initiate the UPR, then by first splicing and new translation.

---

## [Author Response]

*1) The pseudo-polysome phenomenon is nicely addressed experimentally but could be more clearly described in the text. What is observed is A260 that rises about background in a rather amorphous smear – earlier studies that collected more fractions saw the ribosome peaks emerging – this could be discussed more explicitly.*

As suggested, we have now more explicitly discussed the background signal by describing that the *HAC1* signal smears “across all of the translating (i.e., 80*S* and larger) fractions without substantial enrichment in any particular fraction”. In addition we have included two references to the ribosome peaks of *HAC1* signal that were observed in an earlier study.

*To the non-expert, the modest background A260 will not be particularly compelling, and so the modest reduction will not seem compelling either. This is a request for textual changes.*

We have revised the text to highlight the fact that the modest background smear of signal represents a substantial fraction (~50%) of the total mRNA when considered in aggregate and that the reduction is also quite substantial (from ~50% down to <20%).

*2) In all Figures, the 5' and 3' UTR are colored in black as though they are not part of the exons – for clarity, these should be colored in exon color but in thin lines instead of thick (for coding).*

We appreciate the reviewer bringing to our attention this issue with the mRNA schematics. Because we have used similar mRNA schematics for *HAC1* variants as well as for *GFP* and *HIS3* reporter genes, the proposed solution is not ideal, as this would introduce confusion about the 5′-UTR portion of exon 1. In addition, we think it’s helpful to visually distinguish the constitutive 5′ and 3′ UTRs from the variably noncoding regions of the exons. Therefore, we have kept the thin black lines depicting the constitutive 5′ and 3′ UTRs of *HAC1* but have revised the legend of Figure 1 to better explain the mRNA schematics that we use throughout the manuscript. The revised legend now explicitly indicates that ‘constitutive 5′- and 3′-UTRs located within exons 1 and 2, respectively, are shown as thin black lines’ and that ‘the coding regions of exons 1 (teal) and 2 (purple) are labeled as “*HAC1* exon1” and “exon2”, respectively’.

*3) RNA level changes are dismissed as not relevant though in some figures (3A for example), there are substantial changes in RNA level that should not be so quickly dismissed.*

We also noticed that in some cases RNA levels differed among constructs (as might be expected) and did not intend to dismiss these differences as irrelevant. Rather, our intent was to highlight the fact that differences in RNA levels alone cannot explain the differences in protein output that we observe. In two instances in the text, we describe the mRNA abundance data using the phrases “could not be accounted for” and “was not explained by” to explicitly make this point. In the two other instances where we refer to mRNA levels, we have revised the text to better convey this message: When describing Figure 2—figure supplement 1, we now explicitly refer to the comparison of construct 3 with constructs 5–6 as the data supporting the statement that “the mutant mRNAs were present at similar levels compared to the intronless mRNA”; and when referring to Figure 5, we now indicate that mRNAs were similar in abundance when comparing “the presence versus the absence of *DUH1*” rather than when comparing among constructs as originally implied. Importantly, in the figure that the referee mentioned (Figure 3—figure supplement 1), the largest differences in RNA levels corresponded to the constructs expressed from the *TMA7* locus compared to the *HAC1* locus, which is unsurprising given the alternative promoter.

*4) Discussion of why this transcription factor Hac1u needs to be so tightly controlled, but why translational control is not sufficient, was a bit too long and speculative. This portion of the Discussion could be reduced.*

We have removed the most speculative portion of this discussion to make it shorter, while also incorporating the additional suggestion of Reviewer #3.

*Reviewer #2:*

*The experimental work is very convincing and aside from a few minor points about inadequate citation (see below), I have no major concerns. I think this paper will generate wide interest because it raises so many new questions about UPR regulation and, more generally, the breadth of targets of the ubiquitin proteasome system.*

*Concern about adequate citation:*

*The original discovery of Hac1p function by Cox and Walter in 1996 showed that 1) Uninduced Hac1p is an effective stimulator of the UPR and 2) E2 ligases are involved in degrading uninduced Hac1p. Subsequent papers (including most recently work by the Dever and Dey labs) also showed that uninduced Hac1p is capable of conferring resistance to tunicamycin.*

*Therefore, the authors should eliminate the statement that their results, "confirm our hypothesis that Hac1up is indeed a functional UPR transcription factor." This was already established and should be acknowledged as such.*

We apologize for the confusion caused by the poor wording of this statement and have revised the manuscript accordingly. The hypothesis confirmed by our results was that “Duh1p-dependent degradation normally masks the activity of Hac1^u^p”, not that Hac1^u^p is an active transcription factor.

*Second, they should point out that the idea of the intron "tail" sequence serving as a degron dates back two decades. While it's true that this model was not strongly embraced by the field, the Walter lab should be cited for first proposing and testing it.*

As suggested, we have added three additional citations to the Cox & Walter (1996) publication that first proposed that the tail could act as a degron (once each in the Introduction, Results, and Discussion sections).

*Reviewer #3:*

*My comments/questions are therefor of a fairly minor nature, but might also be of interest to other readers.*

*1) In the figures, can the authors unify the schematics for the mRNA structures. Specifically, I thought Figure 1 should have the same colour coding and intron included.*

We meant for Figure 1 to depict an arbitrary mRNA rather than *HAC1*, which is why we chose to use a different color scheme. We have indicated this in the figure legend by referring to “an mRNA” rather than “*HAC1* mRNA”.

*At first reading of Figure 1, it took a while to recognise that fat-to-thin line transition meant inclusion of 3' nonsense stop codons. I'm not sure if this needs more explicit marks on the figures.*

To maintain consistency with other schematics, we did not explicitly mark the nonsense stop codons in Figure 1 but instead described in the legend “resulting ORFs shown as thick colored boxes”.

*Also should the purple line on the left-hand panel of Figure 8 be fat?*

We confirmed that the purple lines in Figure 8 are correctly depicted as thick/thin lines where appropriate.

*2) On the topic of the truncating stop-codons should there be comment about why these don't trigger NMD? Or is the fact that they don't more evidence toward the point that these mRNA are not translated.*

Our preliminary analysis of mRNA abundances suggests that the early stop codons don’t trigger NMD, which would be consistent with the mRNAs being untranslated since NMD requires translation. However, we have not directly measured the impact of NMD disruption (e.g., knockout of Upf1/2/3) on mRNA abundances and, therefore, cannot draw definitive conclusions about their NMD sensitivity.

*3) At the end of the second paragraph of the subsection “An additional silencing mechanism downstream of translation initiation”: prompted by the point about 5'-3' looping being an impediment to "binding or progress of the scanning ribosome" I checked the recent TCP-seq App from Archer et al. Nature 2016 (http://bioapps.erc.monash.edu/TCP/) to see the HAC1 transcript had any interesting regulation by small subunit scanning. However, there is no-data, which is unexpected given HAC1s abundance. But it does suggest that the inhibition precedes binding of even a single SSU necessary for the TCP-seq approach.*

We, too, were excited to inspect *HAC1* in the recent TCP-seq dataset and found that there was no data. However, the TCP-seq data were generated in such a way that 80S-free mRNPs like *HAC1* would have been missed due to the initial pelleting step in the protocol. Therefore, we unfortunately could not use the data to gain further insight into the inhibition of *HAC1*.

*4) In the Discussion: I wondered if an additional purpose or benefit of leaky translation might be that inhibition of proteolysis might be a kinetically quicker way to get a functional transcription factor with which to initiate the UPR, then by first splicing and new translation.*

We appreciate the reviewer’s suggestion and have incorporated this idea into the revised Discussion section.